# SQM2.20: Semiempirical quantum-mechanical scoring function yields DFT-quality protein–ligand binding affinity predictions in minutes

Adam Pecina [1,2], Jindřich Fanfrlík [1,2], Martin Lepšík[1] & Jan Řezáč [1] ✉

Accurate estimation of protein–ligand binding affinity is the cornerstone of computer-aided drug design. We present a universal physics-based scoring function, named SQM2.20, addressing key terms of binding free energy using semiempirical quantum-mechanical computational methods. SQM2.20 incorporates the latest methodological advances while remaining computationally efficient even for systems with thousands of atoms. To validate it rigorously, we have compiled and made available the PL-REX benchmark dataset consisting of high-resolution crystal structures and reliable experimental affinities for ten diverse protein targets. Comparative assessments demonstrate that SQM2.20 outperforms other scoring methods and reaches a level of accuracy similar to much more expensive DFT calculations. In the PL-REX dataset, it achieves excellent correlation with experimental data (average $R^2 = 0.69$) and exhibits consistent performance across all targets. In contrast to DFT, SQM2.20 provides affinity predictions in minutes, making it suitable for practical applications in hit identification or lead optimization.

Reliable estimation of protein–ligand (P–L) binding affinities is the cornerstone of computer-aided drug design (CADD). In its structure-based branch, i.e. with the availability of the three-dimensional structure of the target protein, physics-based atomistic models should offer good answers for good reasons, and if they are robust and accurate enough, they could have a true predictive ability much needed in drug discovery[1,2]. Existing methods span a wide range of complexity from approximate scoring functions (SFs) used in docking to advanced calculations of binding free energies based on molecular dynamics simulations or complex quantum-mechanical calculations[3–6]. Their accuracy is, in general, proportional to their steeply growing computational cost, and the search for an accurate yet also efficient method remains an unsolved challenge.

Our approach to this problem is based on semiempirical quantum-mechanical (SQM) methods of computational chemistry[7,8]. With corrections for non-covalent interactions[9,10], they describe geometries and energetics of molecular complexes better than molecular mechanics (MM) force fields[11], yet they are still applicable to systems with thousands of atoms on a timescale of minutes[12]. Over the last decade, we have been developing SFs based on SQM calculations and applied them successfully to various targets and tasks, which had been reviewed in refs. 7,8. In our previous studies, we used different methods, often tailored to a specific problem or even to specific targets. In some cases, we opted for computationally more demanding quantum-mechanical methods that can not be used on a larger scale. Here, we leverage our experience to formulate a universal SQM-based SF, named SQM2.20, for general use across diverse protein targets, various ligand chemistries and modes of non-covalent interactions. It is free of any empiricism (apart from system-independent parameters in the underlying computational methods) and is neither tuned to a specific target, nor to protein–ligand interactions in general. The SQM2.20 SF covers the most important contributions to P–L binding

[1]Institute of Organic Chemistry and Biochemistry of the Czech Academy of Sciences, Prague, Czech Republic. [2]These authors contributed equally: Adam Pecina, Jindřich Fanfrlík. ✉e-mail: rezac@uochb.cas.cz

free energy, and all these terms have been updated to the best methods available at this computational level. Together, these changes led to a significant improvement over our previous work. Moreover, this was achieved without compromising the excellent computational efficiency. It is, however, still an end-point SF (using a single representative structure of a particular P–L complex) where some terms are neglected and error cancellation plays a role. As a result, it is applicable to the ranking of a series of ligands of a protein, but it does not yield absolute binding free energies (which would require extensive sampling not feasible at the SQM level).

The ultimate test of an SF's performance is its comparison with experimentally determined binding affinities when applied to multiple different target proteins, each with a series of ligands. Because we aim at high accuracy, the requirements on the experimental reference also need to be very strict. The affinities for each ligand series must be consistent, measured using the same method under the same conditions – ideally in a single study or at the same laboratory[13]. Moreover, as we strive to obtain a good answer for a good (structural) reason, reliable experimental geometries of the P–L complexes are needed. Data matching these requirements are rare. Therefore, in this work we compile a unique dataset of reliable experimental structures and affinities, the Protein–Ligand Refined EXperiment (PL-REX) set. It comprises ten diverse protein targets, each with ten to thirty ligands. No ligands from the original sources that met the above criteria were arbitrarily discarded, even if they were difficult to score and negatively affected the final results. Although the PL-REX dataset comprises multiple crystal structures of P–L complexes within each series, we choose a single protein conformation for each target, into which all the other ligands are inserted by overlapping the crystal structures. The protonation states of selected proteins and ligands are meticulously prepared and manually checked for non-trivial issues. Prior to scoring, these complexes are partially optimized (only the ligand and its close surroundings, using an SQM/MM setup) so that the protein can conform to the geometry of each ligand. We offer the resulting geometries to the general public as the PL-REX dataset on GitHub (https://github.com/Honza-R/PL-REX, also archived at Zenodo[14]). These geometries are referred to as 'PL-REX geometries' in the rest of the paper.

The SQM2.20 score is defined as a sum of terms with clearly defined physical meaning:

$$\text{SQM2.20 score} = \Delta E_{\text{int}} + \Delta\Delta G_{\text{solv}} + \Delta G_{\text{conf}}(L) + \Delta G_{H+} - T\Delta S \quad (1)$$

The individual terms stand for gas-phase interaction energy ($\Delta E_{\text{int}}$), the change of solvation free energy upon complex formation ($\Delta\Delta G_{\text{solv}}$), the change of conformational free energy of the ligand in an aqueous environment ($\Delta G_{\text{conf}}(L)$), the free energy of proton transfer between the ligand and the buffer ($\Delta G_{H+}$) and the loss of ligand conformational entropy ($T\Delta S$) upon binding. $\Delta E_{\text{int}}$ is computed at the PM6-D3H4X level[9,10,15] with recently reparameterized corrections[16–18]. In our experience, this method provides the most accurate description of non-covalent interactions in large systems including fragments of P–L complexes[19]. Another reason for choosing PM6-D3H4X is the linear-scaling implementation of PM6 in MOPAC[20], the MOZYME algorithm[12], which provides a significant speedup compared to other SQM methods. $\Delta\Delta G_{\text{solv}}$ is evaluated using the COSMO2 model[21] at the PM6 level and represents the desolvation penalty connected with protein–ligand complex formation. $\Delta G_{\text{conf}}(L)$ is estimated by optimizing the free ligand and evaluated at the PM6-D3H4X/COSMO2 level. $\Delta G_{H+}$ is only considered if the protonation of the ligand changes upon binding (as evidenced by the estimated $pK_a$ values of the ligands and the structures of the complexes) and is computed at the PM6-D3H4X/COSMO2 level. The entropic term, $T\Delta S$, is computed using the empirical model LM5 fitted to SQM calculations[22]. For efficiency, the score is evaluated on a model of ~2000 atoms, comprising all residues within 10 Å around

all the overlaid ligands in each target protein. We have verified that this model perfectly reproduces the computationally more demanding scoring in a whole protein (see the Supplementary Note 6). The entire workflow, from preparing the structures for calculation to evaluating each component of the SQM2.20 score, is outlined in Fig. 1.

The SQM2.20 SF, with an average calculation time of ~20 minutes per P-L complex, is intended to fill the gap between very fast but less accurate SFs used in docking or virtual screening and much more expensive quantum chemical methods such as density functional theory (DFT). We have therefore compared it to a representative panel of top-performing academic and commercial SFs, as identified by the CASF-2016 update benchmarking study[23], and including additional structure-based machine-learning SFs. On top of that, we have added MM-based SFs as well as state-of-the-art DFT calculations.

This work summarizes the construction of the benchmark PL-REX dataset and its application to validation of various SFs for reliable affinity predictions. The results on the PL-REX dataset demonstrate that only SQM2.20 and DFT-based SFs perform consistently well in all the targets; however, SQM2.20 can be computed in minutes as opposed to DFT, which takes hours or days. SQM2.20, as the only universal SF providing accurate affinity predictions rapidly, can be used in hierarchical protocols for refining the results of simpler calculations in the early stages of CADD or as a principal tool in the lead optimization phase.

## Results and discussion
### Reference dataset for protein–ligand scoring
We have constructed the PL-REX benchmark dataset of ten protein targets, each with a series of ligands (164 complexes in total) for which high-quality experimental data are available. The systems and their main characteristics are listed in Table 1; more details are provided in the Supplementary Information (SI), Supplementary Note 2. X-ray crystal structures were available for 147 complexes, and models of seventeen other complexes were built by modifying ligands closely similar to those for which crystal structures were available (following the rules listed in the SI, Supplementary Note 1). The ligands in PL-REX cover a large chemical space and feature linear, branched and

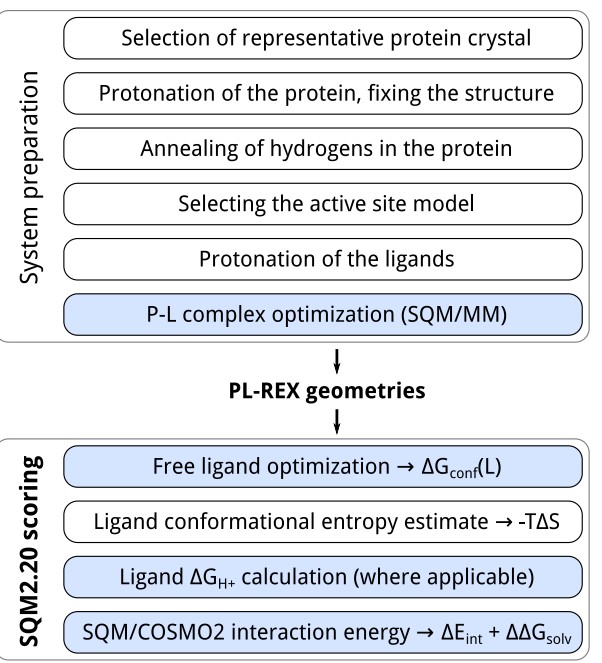

**Fig. 1 | A diagram of the workflow used to prepare the systems for computation and generate the final PL-REX geometries, as well as the SQM2.20 score evaluation itself.** Steps involving SQM calculations are shown in blue.

**Table 1 | Composition and features of the benchmark protein–ligand dataset PL-REX**

| Target | Protein | Ligands | Crystals | pKi range | Ligand similarity[a] | Crystal used, resolution | Notes |
|---|---|---|---|---|---|---|---|
| 01-CA2 | Human carbonic anhydrase II | 10 | 10 | 2.2 | 0.32 | 5NXG, 1.2 Å | – Ligand binding via $Zn^{2+}$ |
| 02-HIV-PR | HIV-1 protease | 22 | 12 | 5.1 | 0.51 | 2AQU, 2.0 Å | – Large flexible ligands<br>– Proton transfer upon ligand binding<br>– One water molecule bridging the protein and ligands |
| 03-CK2 | *Zea mays* casein kinase 2 | 16 | 16 | 1.9 | 0.32 | 3KXN, 2.0 Å | – Halogen bonding<br>– Proton transfer upon ligand binding |
| 04-AR | Human aldose reductase | 14 | 14 | 2.8 | 0.47 | 4XZH, 1.0 Å | – NADP cofactor<br>– Halogen bonding |
| 05-Cath-D | Human cathepsin D | 10 | 3 | 3.5 | 0.71 | 6QCB, 1.55 Å | – Macrocyclic inhibitors |
| 06-BACE1 | Human beta-secretase 1 | 16 | 16 | 3.6 | 0.48 | 5QCZ, 2.3 Å | – Acyclic and macrocyclic inhibitors<br>– One water molecule bridging the protein and ligands |
| 07-JAK1 | Human Janus kinase 1 | 12 | 12 | 3.4 | 0.55 | 4IVD, 1.93 Å | – Six water molecules bridging the protein and ligands |
| 08-Trypsin | Bovine trypsin | 15 | 15 | 4.4 | 0.45 | 1K1I, 2.2 Å | – Proton transfer upon ligand binding<br>– Ten explicit water molecules |
| 09-CDK2 | Human cyclin-dependent kinase 2 | 31 | 31 | 3.6 | 0.65 | 3R9H, 2.1 Å | – Flexible glycine-rich loop covering the binding site |
| 10-MMP12 | Human matrix metallopeptidase 12 | 18 | 18 | 3.9 | 0.47 | 3EHY, 1.9 Å | – Ligand binding via $Zn^{2+}$<br>– Proton transfer upon ligand binding |

[a]The ligand similarity is expressed as the average of the Tanimoto coefficients computed for each pair of ligands.

macrocyclic inhibitors. The total charges of the ligands range from −1 to +2, and the number of rotatable bonds ranges from 0 to 25.

A single representative protein structure per target was selected for scoring based on the criterion that it could best accommodate all the ligands (after the exclusion of incomplete protein structures). First, a single protonation state of the selected protein structure was determined with respect to the prior literature, experimental conditions and all the ligands in the series. Second, the protonation of ionizable groups in the ligands was solved and corrected manually according to the experimental conditions, $pK_a$ calculations, the literature and, most importantly, adjusted to match the selected protein structure. Non-trivial issues, i.e. the protonation states of 20 ligands altered upon binding (a proton is released in 11 cases and taken up in 9 cases) are listed in the SI, Supplementary Note 2.

**SQM2.20 scoring**
We used the SQM2.20 SF at the PM6-D3H4X/COSMO2 level (Eq. 1) for the scoring of the PL-REX dataset. We then evaluated the performance of the SF by examining the correlation between the resulting scores and binding free energies ($\Delta G_{bind}^{0}$) derived from experimental affinities and quantifying it in terms of the squared Pearson coefficient, $R^2$ (Fig. 2, Supplementary Fig. 1 and Supplementary Table 1). SQM2.20 achieved excellent performance across the whole dataset (average $R^2 = 0.69$) with no system failing to obtain a good correlation (the minimal correlation of $R^2 = 0.56$ obtained for 07-JAK1 is still better than the average performance of standard SFs, as detailed below). A notable achievement is the ability to reliably rank not only structurally diverse ligands, but even ligands with different molecular charges (found in 5 out of the 10 series).

The effect of using SQM calculations was assessed by stepwise replacing the geometries and energy terms of the scoring protocol with their MM equivalents (using AMBER ff19SB/GAFF2 force fields[24,25] with IGB7 implicit solvent[26]). First, SQM2.20 scores calculated on MM-optimized geometries resulted in a significant drop of correlation ($R^2 < 0.36$) in three targets, while the average $R^2$ dropped to 0.52 (Fig. 2). This highlights the importance of the quality of the input geometries featured in the PL-REX dataset. Second, the MM scores (with $\Delta E_{int}$, $\Delta\Delta G_{solv}$ and $\Delta G_{conf}(L)$ in Eq. 1 computed at the MM level) calculated on MM-optimized geometries resulted in a dramatic

deterioration of the average $R^2$ to 0.23 (Fig. 2). This is an important finding suggesting that even a rather simple end-point scoring function benefits from accurate SQM calculations more than it would from covering additional contributions but staying at the MM level.

These results motivated us to explore the effects of replacing the SQM method used for the calculation of the leading term, $\Delta E_{int}$, in SQM2.20 with the state-of-the-art DFT benchmark using an accurate but also computationally much more demanding setup (range-separated hybrid ωB97X-D3BJ functional / DZVP-DFT basis set; see the Methods section). To make these calculations feasible, we had to trim the systems to about 1,000 atoms (from ~2,000 atoms). The results obtained with these trimmed models using SQM2.20 still correlated well with experimental data (average $R^2 = 0.62$, Fig. 2) but at the level of individual targets, the correlation deteriorated significantly in 06-BACE1 ($R^2 = 0.37$). In this target, the truncation of the active site model results in an unphysically large molecular charge ($+5$), which is only compensated by the inclusion of more distant anionic amino acid residues in the larger model. This shows the importance of including larger protein surroundings of their ligands for consistently excellent performance, as is the case with the default model. The DFT scoring was evaluated in eight targets (excluding 06-BACE1 for structural reasons already demonstrated at the SQM level and 04-AR where DFT failed to converge in multiple iodine-containing ligands). In this subset, SQM2.20 with the trimmed models reached an average $R^2$ of 0.67, and the DFT scoring yielded a very similar result with an average $R^2$ of 0.64 (Fig. 2). The equivalence of SQM and DFT interaction energies ($\Delta E_{int}$) was confirmed also by comparing them across the whole dataset, which yielded a correlation with an $R^2$ of 0.98 (Supplementary Fig. 2). To summarize, SQM2.20 yielded similar accuracy as the benchmark DFT scoring, but it was four orders of magnitude faster (see Supplementary Note 4 and Supplementary Table 2 in the SI).

**Performance of standard SFs in the PL-REX dataset**
The performance of SQM2.20 scoring in the PL-REX dataset was compared with that of eighteen standard SFs and four structure-based machine-learning (ML) methods. To make the comparison as fair as possible, the scoring was performed on PL-REX geometries as featured in the PL-REX dataset (see Methods). The correlations ($R^2$) averaged over all PL-REX targets of all the standard and ML methods ranged

| Target | Default model (~ 2,000 atoms) | | | Trimmed model (~ 1,000 atoms) | |
|---|---|---|---|---|---|
| | SQM2.20 | SQM2.20// AMBER | AMBER SF | SQM2.20 | DFT score |
| 01-CA2 | 0.67 | 0.36 | 0.28 | 0.63 | 0.85 |
| 02-HIV-PR | 0.75 | 0.70 | 0.33 | 0.71 | 0.61 |
| 03-CK2 | 0.81 | 0.70 | 0.17 | 0.63 | 0.53[b] |
| 04-AR | 0.70 | 0.56 | 0.01 | 0.60 | n.d. |
| 05-Cath-D | 0.66 | 0.22 | 0.23 | 0.70 | 0.66 |
| 06-BACE1 | 0.63 | 0.57 | 0.37 | 0.37 | 0.25 |
| 07-JAK1 | 0.56 | 0.57 | 0.03 | 0.59 | 0.49 |
| 08-Trypsin | 0.75 | 0.73 | 0.54 | 0.61 | 0.79 |
| 09-CDK2 | 0.61 | 0.20 | 0.07[c] | 0.56 | 0.50 |
| 10-MMP12 | 0.74 | 0.62 | 0.03[c] | 0.81 | 0.69 |
| Average | 0.69 | 0.52 | 0.23 | 0.62 (0.67[a]) | 0.64[a] |

[a] Average $R^2$ excluding 04-AR and 06-BACE1 systems;
[b] $R^2$ excluding iodine-containing 3KXN ligand;
[c] Correlations with a negative value of $R$;
n.d., not determined

**Fig. 2 | Squared Pearson coefficient ($R^2$) of the correlation between the SQM2.20 SF and its MM and DFT derivatives on the one hand and experimentally obtained binding free energies, $\Delta G_{bind}^0$, on the other, calculated for the PL-REX dataset.** SQM2.20//AMBER stands for SQM2.20 for scoring on AMBER (ff19SB/GAFF2 force fields and IGB7 implicit solvation model, see the Methods section) geometries obtained via optimization of the ligands and their close surroundings. AMBER SF signifies that the ff19SB/GAFF2 force field and IGB7 implicit solvation model were used for geometry optimization of the ligands and their close surroundings as well as scoring. DFT score means that $\Delta E_{int}$ was calculated at the $\omega$B97X-D3BJ/DZVP DFT level (see the Methods section). Note that differences in $R^2$ less than 0.1 are deemed not significant. Source data are provided as a Source Data file.

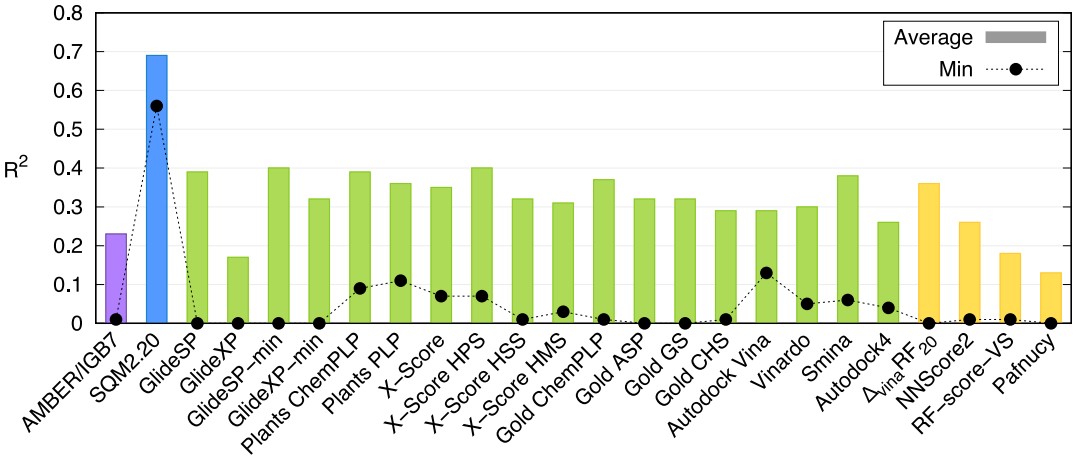

**Fig. 3 | Average (columns) and minimal (black circles) correlations ($R^2$) over the PL-REX dataset.** SQM2.20 in blue; AMBER/IGB7 in purple; standard SFs in green; ML methods in orange. Source data are provided as a Source Data file.

from 0.13 to 0.40 as compared to the range from 0.56 to 0.81 of SQM2.20 (Fig. 3). The best SFs were GlideSP-min and X-Score HPS with an average $R^2$ of 0.40, but this was still well below the lowest correlation achieved by SQM2.20 ($R^2 = 0.56$). We also tested the performance of the standard SFs on MM-optimized geometries. The results were very similar (an average $R^2$ also ranged from 0.13 to 0.40; the best being PLANTS ChemPLP SF; see details in the SI, Supplementary Note 3, Supplementary Fig. 3 and Supplementary Table 3), which shows a smaller dependence of the classical SFs on the initial geometries.

Analysis over the individual targets found that SQM2.20 was the only universal SF, meaning that it was fully consistent across all ten targets, as indicated by the smallest standard deviation of the average $R^2$ of 0.69 ± 0.08. In comparison, the standard SFs and ML methods reached considerably worse or even no correlation with experimental data for at least some targets (see Supplementary Table 3), with the best results found for GlideSP-min (average $R^2 = 0.40 \pm 0.34$; $R^2 < 0.5$ in five targets), and X-Score HPS (average $R^2 = 0.40 \pm 0.26$; $R^2 < 0.5$ in five targets), followed by Smina (average $R^2 = 0.38 \pm 0.27$; $R^2 < 0.5$ in five targets) and the $\Delta_{vina}RF_{20}$ ML-based approach (average $R^2 = 0.36 \pm 0.28$; $R^2 < 0.5$ in five targets). The remaining SFs yielded poor levels of correlation ($R^2 < 0.5$) in more than half of the ten targets. Here we found one result worth noting – the more sophisticated GlideXP scoring function did not outperform the GlideSP; however, this is consistent with an earlier benchmark study[23]. On the other hand, in seven of the ten targets of PL-REX, there was at least one SF with a reasonable level of correlation ($R^2 > 0.50$), which justifies their use in specific systems, provided that their performance is validated beforehand.

These results can also be used to identify challenging targets in the PL-REX dataset. The most difficult ones were 01-CA2, 03-CK2 and

**Table 2 | Average performance (squared Pearson correlation coefficient, $R^2$, with respect to $\Delta G_{bind}^0$) of 22 standard SFs and ML methods applied to PL-REX geometries of the PL-REX dataset**

| Target | $R^2$, avg. ± st.dev. | Best $R^2$ (best SF) | SFs with $R^2 > 0.50$ |
|---|---|---|---|
| 01-CA2 | 0.12 ± 0.11 | 0.34 (Gold ASP) | 0 % |
| 02-HIV-PR | 0.13 ± 0.14 | 0.58 ($\Delta_{vina}RF_{20}$) | 5 % |
| 03-CK2 | 0.14 ± 0.14 | 0.47 (RF-score-VS) | 0 % |
| 04-AR | 0.53 ± 0.22 | 0.84 (Gold GS) | 73 % |
| 05-Cath-D | 0.51 ± 0.20 | 0.80 (PLANTS ChemPLP) | 64 % |
| 06-BACE1 | 0.42 ± 0.23 | 0.76 (X-Score) | 36 % |
| 07-JAK1 | 0.47 ± 0.22 | 0.77 (Gold GS) | 55 % |
| 08-Trypsin | 0.49 ± 0.22 | 0.85 (Gold ASP) | 55 % |
| 09-CDK2 | 0.15 ± 0.06 | 0.25 (GlideSP) | 0 % |
| 10-MMP12 | 0.16 ± 0.12 | 0.57 (NNScore2) | 5 % |

Source data are provided as a Source Data file.

09-CDK2, where all standard SFs and ML approaches failed completely ($R^2 = 0.12 ± 0.11$, $0.14 ± 0.14$ and $0.15 ± 0.06$, respectively, see Table 2). Here, 01-CA2 is a zinc metalloprotein and 03-CK2 features challenging halogen bonds; we have not, however, found any obvious explanation for the failure in 09-CDK2. In two more targets, 02-HIV-PR (featuring very large ligands) and 10-MMP12 (zinc metalloprotein), the standard and ML methods failed (average $R^2 = 0.13 ± 0.14$ and $0.16 ± 0.12$), with the sole exception of $\Delta_{vina}RF_{20}$ ($R^2 = 0.58$) in the former case and of NNScore2 ($R^2 = 0.57$) in the latter. The SQM2.20 SF works well in all these challenging cases. In the remaining five targets (04-AR, 05-Cath-D, 06-BACE1, 07-JAK1 and 08-Trypsin), both SQM2.20 and many of the standard approaches performed well ($R^2 \geqslant 0.5$). Nevertheless, only three standard SFs showed consistent performance across these five targets similar to the SQM2.20 SF (average $R^2 = 0.66 ± 0.07$): Glide's SP with and without refinement ($R^2 = 0.72 ± 0.10$ and $0.66 ± 0.09$, respectively) and X-Score HPS ($R^2 = 0.63 ± 0.09$).

This comparison reveals the strong point of SQM-based scoring. It is able to deal with diverse P–L binding motifs without any system-specific parameters and prior data. SQM calculations treat all P–L interactions on equal footing, capturing their physics as accurately as possible at this computational level. This approach increases the chance of success when working with novel targets, ligands with unusual chemistry or in other challenging cases.

Conventional SFs and ML-based approaches are of course much faster (taking mere seconds to complete), and SQM scoring with an average computational time of about 20 minutes is not their direct replacement (for details on the timing, see Supplementary Note 6, Supplementary Table 4 and Supplementary Fig. 4 in the SI). However, SQM scoring can be used to refine results of conventional SFs when greater accuracy is needed. Moreover, it is clear that there are cases where the SQM scoring is the most efficient approach that brings useful results, and this fact itself should make it a tool to be considered in practical CADD applications.

This study answers several fundamental questions of computational drug design: on the existence of a universal yet computationally efficient physics-based scoring method, on the real performance of scoring methods when applied to known structures for which high-quality experimental data are available, and on the necessity to invoke a higher level of theory and at what cost.

We have developed the universal scoring function SQM2.20 based on semiempirical quantum-mechanical calculations. It is a purely physics-based end-point SF addressing the leading terms of protein–ligand binding free energy, utilizing the most advanced and highly efficient computational methods available. It does not use any empirical parameters tailored either to a specific target or

protein–ligand binding in general, yet it describes the binding energetics in P–L complexes quantitatively.

To validate the SQM2.20 scoring function, we have assembled the benchmark dataset PL-REX consisting of carefully curated experimental data – specifically, ten varied protein targets with at least ten inhibitors each, featuring high-resolution crystal structures and consistently measured affinities. This gives us confidence that the computational results we obtain have a solid structural basis. The PL-REX dataset is available to the general public and is intended as a tool for the rigorous validation of existing and newly developed SFs.

Our comparative assessment of academic and commercial standard and ML scoring methods identified SQM2.20 as the only SF producing high-quality ranking of ligands across the diverse P–L series featured in the PL-REX dataset. SQM2.20 achieved a very good level of correlation with experimentally determined $\Delta G_{bind}^0$, with an average $R^2$ of 0.69 and the lowest $R^2$ of 0.56. The average correlations achieved by the other SFs did not exceed the $R^2$ value of 0.40, and none of them was able to describe all the targets with consistent quality. Our results also demonstrate that in an identical workflow, the SQM method provides a significant advantage over an MM force field, but switching from SQM to much more expensive DFT calculations does not improve the results further.

As such, the SQM2.20 SF fills the gap between ultrafast standard SFs (taking seconds to compute) and costly DFT methods (taking days), offering accurate results on a timescale of minutes. This makes it attractive for practical applications in middle-stage refinement and in lead optimization phases of structure-based CADD workflows. To facilitate this, we plan to implement the SQM2.20 scoring function in a standalone tool that would automate the entire workflow; these developments will be announced separately in the future.

## Methods
### PL-REX dataset
The dataset was constructed from P–L complexes with consistently obtained binding affinities and relevant structural data, specifically $K_d$, $K_i$ or IC$_{50}$ measured ideally at one laboratory under the same conditions and X-ray crystal structures of the respective P–L complexes from the Protein Data Bank (www.rcsb.org). For crystal structure quality metrics, see Supplementary Table 7. Ligands without a crystal structure (17 out of 164) were considered only in cases where they could be modeled with confidence based on the crystal structure featuring a similar ligand. Additionally, we require the p$K_i$ ($-\log K_i$) range to be greater than 1.5 and the number of ligands to be at least 10.

For the majority of the complexes, there are available experimental dissociation constants ($K_d$) or inhibition constants ($K_i$) which we treat as being equivalent under the assumption of a competitive mechanism of inhibition. In the remaining cases, we use experimental IC$_{50}$ values to approximate the $K_i$ as IC$_{50}$/2, assuming that the concentration of the substrate in the experiment is close to the Michaelis-Menten constant. Even if this approximation was not accurate, it is a linear relationship that would not affect the correlation between scores and the experiment, as long as all the IC$_{50}$ values come from a consistent series of experiments under the same conditions, which we verified in the original sources. Finally, we convert the experimental $K_d$, $K_i$ or its estimate to a free energy of binding ($\Delta G_{bind}^0$) that is compared to the calculations. It should be kept in mind that in the case of the IC$_{50}$ values, it is only an estimate, but because this approximation does not affect the final results, we do not mention it explicitly in the remainder of the paper.

For each target, a single representative protein conformation was selected. For 9 out of 10 targets, we systematically chose the protein structure that could best spatially accommodate all the aligned ligands. The closest atom-atom distance between each protein

structure and all the ligands was measured as an indication of the ability to accommodate the ligands, and the protein with the largest measured value was selected. Protein crystals with missing side chains in the binding sites were discarded prior to the selection. Only in the case of the 05-Cath-D target, where only 3 crystal structures are available and 7 ligands were modeled, we selected the native protein structure of a ligand that was the most similar to the manually built compounds. To evaluate how the choice of the receptor structure affects the results, SQM2.20 was also tested on protein structures selected according to different criteria. In addition, in the case of the 01-CA2 target, we also scored each ligand in its own native crystal structure. Both these analyses are discussed in the SI, Supplementary Note 7 and Supplementary Table 5).

All non-standard residues (except the cofactor in 04-AR), ions (except $Zn^{2+}$ in 01-CA2 and $Zn^{2+}$ and $Ca^{2+}$ in 010-MMP12) and co-solvents were discarded. Water molecules were retained only in cases where they defined the binding mode of the ligand (forming networks in the protein structure or bridging the protein and the ligands). As a result, explicit water molecules were included and treated as part of the protein in the cases of four proteins (see Table 1). Hydrogens and missing heavy atoms were added to the protein using the Leap tool of AMBER 20 suite[27]. Protonation of ionizable amino acid side chains (e.g. histidines and aspartates) in the binding sites were checked visually with respect to the crystallographic conditions, surroundings and literature. By default, alternative residue conformations A were considered unless there were specific reasons for other conformations. Hydrogens were added to the ligands by using the software Obabel v. 2.3 (http://openbabel.org)[28]. Protonations of ionizable groups in the ligands were checked manually and corrected according to the experimental conditions, surroundings, $pK_a$ calculations (Stardrop v. 7.0, https://optibrium.com/stardrop) and the literature. 2D structures of all ligands, as well as schematic drawings of the binding mode in the representative P-L complexes, are provided in the Supplementary Note 10 and Supplementary Figs. 32–51 of the SI.

For each target, the default models were defined as residues of the representative protein within 10 Å of all the overlaid ligands and amounted to 1,295–2,298 atoms (a residue is selected if at least one of its atoms fits within the cutoff distance). The trimmed models comprised 6 Å surroundings (only in the cases of 02-HIV-PR and 06-BACE1 it was reduced to 5 Å because of the large size of the ligands) of all the overlaid ligands and were made up of 753–1,096 atoms. The protein backbone was then terminated with neutral caps of the peptide bond that introduce the least perturbation into the system ($-NH-CH_3$ at the C-terminus and $-CH=O$ at the N-terminus). Truncating the active site model could expose charged amino acid residues at the boundary; we did not neutralize these as they were far from the ligands and were effectively screened by the solvent model. The P–L models were gradually relaxed in a series of optimizations. First, we performed annealing (60 ps molecular dynamics with a thermostat cooling of the system from 300 to 0 K) and optimization of the hydrogen atoms in the protein only. Second, we formed the P-L complexes, and possible clashes with the protein were resolved by local geometry optimizations at the MM level. The final step was a free gradient optimization of the ligand and its surroundings (all protein residues within 4 Å of all the ligands) in an aqueous environment using a hybrid SQM/MM approach where the SQM part (treated with PM6-D3H4X) comprised only the ligand. The optimizations were performed in Cuby4 (http://cuby4.molecular.cz)[29], which also provided the interface implementing the SQM/MM scheme. The final structures used for the scoring are available in a public repository (https://github.com/Honza-R/PL-REX) and are referred to as PL-REX geometries.

## Scoring setup
The PL-REX dataset was scored using the SQM2.20 SF as well as its MM (AMBER SF) and DFT derivatives alongside with eighteen standard SFs

widely used in commercial or academic settings and four machine-learning approaches. Besides the PL-REX geometries (Fig. 3; Supplementary Table 3A), we also evaluated the standard SFs against locally optimized ligand poses (using AMBER ff19SB/GAFF2 force fields[24,25] with IGB7 implicit solvent[26]) in the rigid protein structure, referred to as "MM-opt LIG" geometries (Supplementary Table 3B), as commonly used in the benchmarking of standard SFs[23]. Neither SQM2.20 nor most of the other scoring functions tested yield absolute binding free energies for which error values (in comparison to experimental data) can be expressed in energy units (the relationship between the experimental $\Delta G_{bind}^0$ and the SQM2.20 score is discussed in more detail in the SI, Supplementary Note 8, Supplementary Table 6 and Supplementary Fig. 5). We therefore evaluate the performance of the scoring functions in terms of the correlation of the scores they produce with binding free energies derived from experiments, reporting the squared Pearson correlation coefficient ($R^2$) for each target. This captures the ability of an SF to rank different ligands of the same protein, which is the crucial metric for practical applications. To summarize the results for the whole PL-REX dataset, average $R^2$ across the ten targets is used.

### SQM2.20 SF
The SQM2.20 score is computed as a sum of the terms in Eq. 1 computed at the PM6-D3H4X/COSMO2 level[9,10,15,21]. COSMO2 is an extension and reparameterization of the original COSMO implicit solvent model[21]. The PM6 calculations were carried out using MOPAC2016 (http://openmopac.net)[20] with MOZYME[12], and the D3H4X corrections were added using Cuby4 (http://cuby4.molecular.cz)[29]. The latest parameterizations of the D3H4X corrections were utilized[16–18], which is simplified by interfacing the customizable implementation of the corrections in Cuby4 to an unmodified version of MOPAC. Similarly, the Cuby4 interface to MOPAC automates the setup and evaluation of the COSMO2 solvation free energy. When the default PM6-D3H4X/COSMO setup of MOPAC was utilized, the average correlation decreased by 0.13, which was caused mainly by the significant drop of correlation in halogen-bonded 03-CK2 inhibitor series (Supplementary Table 1). Also, the transition from COSMO to COSMO2 significantly improves the description of 01-CA2, a zinc metalloprotein, as it remedies an issue we previously observed in this class of systems[21]. We also note that PM7/COSMO[30] resulted in a slightly decreased average correlation (by 0.10) and that it failed to produce reasonable correlation in three targets (03-CK2, 09-CDK2, 10-MMP12; see Supplementary Table 1). This is not a surprising result, since PM7 tends to overestimate interaction energy in large systems in general[19,31], which also affects its ability to describe P−L complexes[19]. The individual results of the SF variants based on the default PM6-D3H4X/COSMO and on PM7/COSMO calculations are also plotted in the SI (see the Supplementary Note 9 and Supplementary Figs. 30 and 31).

The conformational entropy term $-T\Delta S$ is computed using the empirical LM5 model[22], which estimates the entropy from several descriptors characterizing the ligand and has been fitted to a large database of GFN2-xTB calculations on drug-like molecules. We use the model as implemented and parameterized by its authors. In the present dataset, omitting the $-T\Delta S$ term leads to practically the same overall correlation of the score with the experiment. However, by applying a simpler model which we used previously[8] (a penalty of 1 kcal/mol per rotatable bond), the results deteriorate slightly to an average $R^2$ of 0.63.

In cases where the protonation state of a ligand differed between the unbound and bound forms, the variant from the complex was considered in the calculation of $\Delta E_{int}$, $\Delta\Delta G_{solv}$ and $\Delta G_{conf}(L)$. $\Delta G_{H+}$ was then evaluated based on differences in PM6-D3H4X/COSMO2 energies of protonated and deprotonated forms of the free ligand. Depending on whether the ligand (L) was deprotonated (Eqs. 2, 4) or protonated (Eqs. 3, 5) in solution, the following acid-base

reactions were considered

$$L + H_2O \rightarrow LH^+ + OH^- \qquad (2)$$

$$LH + H_2O \rightarrow L^- + H_3O^+ \qquad (3)$$

and, consequently, the $\Delta G_{H+}$ term is evaluated as

$$\Delta G_{H+} = \frac{10^{(pH-pKa)}}{1+10^{(pH-pKa)}} \cdot (G(LH^+) - G(L) + G(OH^-) - G(H_2O)) \qquad (4)$$

$$\Delta G_{H+} = 1 - \frac{10^{(pH-pKa)}}{1+10^{(pH-pKa)}} \cdot (G(L^-) - G(LH) + G(H_3O^+) - G(H_2O)) \qquad (5)$$

G(L), G(LH), G(LH⁺) and G(L⁻) are computed at the PM6-D3H4X/COSMO2 level (here we use the label G for a quantity that is the sum of the PM6-D3H4X enthalpy of formation in the gas phase and the COSMO2 free energy of solvation). Since the solvation free energy of OH⁻ and H₃O⁺ is difficult to calculate reliably with SQM methods, the G(H₃O⁺), G(OH⁻) and G(H₂O) are computed as the sum of the PM6-D3H4 gas phase enthalpy and the experimental solvation free energy[32]. The fraction of differently protonated species was estimated from the *pH* at which experimental affinities were measured and from the p$K_a$ of the titratable group of the ligand using the rearranged Henderson−Hasselbalch equation. The p$K_a$ was either known from experiments (08-Trypsin ligands, 10-MMP12 ligand fragments) or computed using Stardrop v. 7.0 (02-HIV-PR and 03-CK2 ligands). $\Delta G_{H+}$ was not evaluated for the ligands of 01-CA2, where all the ligands undergo deprotonation of the same functional group, so this term does not affect their relative binding energies. Further, the effect of omitting the $\Delta G_{H+}$ term from SQM2.20 was assessed. The correlation deteriorated moderately in the 08-Trypsin and 10-MMP12 targets ($R^2$ decreased by 0.11 and 0.13, respectively) or negligibly in the 02-HIV-PR and 03-CK2 targets ($R^2$ decreased by 0.03 and 0.06, respectively). In cases where several conformations of a single ligand were present in the crystal structure, all were computed and the most stable one was selected based on the SQM energy of the complex.

### DFT scoring
The DFT score is based on the SQM2.20 SF, in which the $\Delta E_{int}$ term is replaced with DFT calculations. We use ωB97X-D3BJ[33,34], a range-separated dispersion-corrected hybrid DFT functional which provides state-of-the-art description of non-covalent interactions[35]. The parameters in the D3 correction were reoptimized for use with the DZVP-DFT basis set (available through the Basis Set Exchange repository, https://www.basissetexchange.org, as dgauss-DZVP) using the procedure and reference data described in ref. 36. The resulting values of the parameters are $s_8 = 0.9220$, $a_1 = 0.3419$ and $a_2 = 5.2955$. It has been shown that this setup yields interaction energies on par with calculations with a triple-ζ basis set but at a fraction of the computational cost[36]. We used the RI-COSX approximation for acceleration. Despite several attempts, it was not possible to achieve reliable convergence with the iodine-containing ligands of 03-CK2 and 04-AR. The calculations were performed in Orca v. 5.0.3[37].

### AMBER scoring
Amber score refers to the MM analog of SQM2.20 where the $\Delta E_{int}$, $\Delta\Delta G_{solv}$ and $\Delta G_{conf}(L)$ terms of Eq. 1 were evaluated at the MM level. We used the AMBER ff19SB force field[24] for the protein and GAFF2 force field[25] with partial charges extracted from PM6 calculations for the ligands; the environment was described by the IGB7 implicit solvent model[26]. AMBER SF was used for the scoring of P−L complexes

which were partially optimized (ligands and their close surroundings defined in the same way as in the optimization of the PL-REX structures) at the MM level (Fig. 2), of PL-REX geometries (Fig. 3 and Supplementary Table 3A) and of locally optimized ligand poses ("MM-opt LIG" using AMBER ff19SB/GAFF2[24,25] with IGB7[26]) in the rigid protein structure (Supplementary Table 3B).

### Standard SFs
Several empirical, regression-based SFs were used with different terms to describe van der Waals contacts, lipophilic surface coverage, hydrogen bonding, ligand strain, desolvation or metal interaction. These were GlideScore XP (GlideXP) and GlideScore SP (GlideSP), optionally with pose refinement (GlideXP-min and GlideSP-min, respectively), PLANTS PLP (PLP) and ChemPLP (ChemPLP), Auto-Dock4, Autodock Vina, Chemscore of GOLD (CHS), Goldscore (GS) and ChemPLP, HPScore of X-SCORE, HMScore, HSScore and averaged X-Score, Vinardo, and Smina. Knowledge-based potential was represented by Astex Statistical Potential of GOLD (ASP). Default as well as recommended protocols (best practices) were tested for all the methods and setups, and those giving the best correlations are reported. Individual active sites of the targets were defined as boxes centered on the center-of-mass of all overlayed ligands of the particular P−L series. The scores obtained with these methods are available in the PL-REX repository, and their plots against the experimental results are provided in the SI as Supplementary Figs. 8–21.

### Glide
We used four modes of the internal empirical scoring function of the Glide module (v. 9.4.141, mmshare v. 5.7.141) in the software Schrö-dinger (v. 2022-1), namely Standard Precision (SP) and Extra Precision (XP)[38,39], each with and without minimization of the ligand structure inside the receptor (referred to as GlideSP-min, GlideXP-min, GlideSP and GlideXP scores, respectively). The dimension of each side of the inner grid box was set to 20 Å. The 2+ charges of Zn and Ca ions were corrected manually. Halogens and aromatic hydrogens were defined as potential H-bond donors. For multiple P−L complexes of 01-CA2 and one complex of 10-MMP12, scoring by GlideSP and GlideXP without minimization resulted in unphysically large positive final binding scores (with values of 10,000).

### PLANTS
Two empirical SFs of the Protein−Ligand ANT System (PLANTS) v. 1.2. were used: PLP and ChemPLP[40]. Both these SFs use piecewise linear potential (PLP) to model the steric complementarity of the ligand and the protein. The ChemPLP SF is based on Chemscore of GOLD and introduces angle-dependent terms for H-bonding and metal binding. The protein and ligand molecules were loaded in Mol2 format and processed by SPORES, a structure protonation and recognition tool. The binding site radius was set to 12 Å and the scoring was run with the default setup using simplex optimization mode.

### X-Score
We used three individual empirical SFs of the X-Score v1.2 package (i.e. HPScore, HMScore and HSScore)[41], combining terms for van der Waals interactions, hydrogen bonding as well as hydrophobic and deformation effects. These three variants only differ in the algorithm for calculating the hydrophobic effect term. All three neglect all water molecules. X-Score is defined as the average of HPScore, HMScore and HSScore. The protein molecules were loaded into X-Score software in PDB format whereas the ligand molecules were loaded in Mol2 format. The input structures were processed with the utilities 'FixPDB' and 'FixMOL2'. All other parameters were set to their default values during the scoring process. The resulting scores in p$K_d$ units were converted to $\Delta G$ and examined for correlation with $\Delta G_{bind}^0$.

## GOLD

In the GOLD Suite (v. 2022.1.0, Cambridge Crystallographic Data Center)[42], we used three empirical scoring functions (ChemPLP, GS and CHS) and one knowledge-based function (ASP) based on atom–atom distance potential added to some Chemscore terms. The protein and ligand molecules were loaded in Mol2 format. The binding site radius was set to 12 Å. Zinc coordination geometries were set to tetrahedral for Zn-containing proteins. We used simplex minimization during scoring in GOLD and kept all the other settings at their default values. The resulting scores with a negative sign were examined for correlation with $\Delta G_{bind}^0$. None of GOLD SFs were able to score the 3KXG complex of 03-CK2 series (not included in the final statistics).

## AutoDock

We used two engines of the Autodock Suite (v. 4.2.6) with three SFs, namely Autodock 4 SF, Autodock Vina SF v. 1.2.0[43–45] and Vina Radii Optimized SF (Vinardo)[46]. Protein molecules were loaded in PDB format and ligand molecules in PDBQT format; they were then processed by the Ligand4.py and prepare_receptor.py scripts in AutoDockTools-1.5.6 with default settings. Affinity grid maps were calculated within a 20 Å box. In Autodock 4, tetrahedral zinc pseudo-atoms around zinc ions were used together with the AutoDock4Zn improved force field in the case of metalloproteins (01-CA2 and 10-MMP12)[47].

## Smina

We used the DKoes_scoring built-in SF of the Smina fork of AutoDock Vina (denoted here as Smina SF)[48]. Smina SF uses a 4–8 Lennard-Jones potential with terms combined from Vina and Autodock 4, such as Vina's term for H-bond interactions and the Autodock 4 term for solvation effects. Input structures were handled in the same way as for Autodock Vina and a default setup with a 200-step minimization was used for scoring.

## Machine-learning approaches

In CADD, machine learning methods are used mainly in ligand-based approaches, and only a few structure-based ML methods are available. These are either standard SFs extended with additional ML correction ($\Delta_{vina}RF_{20}$, NNScore2.0) or standalone ML algorithms estimating affinity directly from the structure of the complex (RF-score-VS, Pafnucy). The obtained absolute dissociation constants ($pK_d$) were converted to $\Delta G$ and correlated to $\Delta G_{bind}^0$.

## $\Delta_{vina}RF_{20}$

This descriptor-based ML model uses the score computed by Auto-Dock Vina SF and combines it with a random forest-based correction term[49]. The protein molecules were loaded in PDB format, whereas the ligand molecules were loaded in Mol2 format.

## NNScore2.0

The neural network-based scoring function NNScore 2.0 was trained on a large number of P–L binding descriptors derived from Vina1.1.2 and BINANA[50]. The input protein molecules as well as ligand structures were loaded in PDBQT format.

## RF-score-VS

The random forest-based scoring function RF-Score-VS was used in its second version (v2)[51–53]. The protein molecules were loaded in PDB format and the ligand molecules were loaded in Mol2 format. RF-Score-VS was optimized for virtual screening using a random forest algorithm trained on 900,000 docked molecules across 102 targets.

## Pafnucy

Pafnucy is a deep three-layer convolutional neural network[54]. The network was trained and tested on protein–ligand complexes from the PDBbind database version 2016[55] and the Astex Diverse Set[56]. We used

the source code and model parameters from http://gitlab.com/cheminfIBB/pafnucy. Preparatory scripts were used to process the input protein and ligand structures in MOL2 format.

## Reporting summary

Further information on research design is available in the Nature Portfolio Reporting Summary linked to this article.

## Data availability

The prepared and optimized structures of the protein-ligand complexes, the PL-REX dataset, as well as other structures used in the calculations reported in the paper, and the resulting scores generated in this study have been deposited in a GitHub repository https://github.com/Honza-R/PL-REX and are also archived at Zenodo with https://doi.org/10.5281/zenodo.8182922[14]. The crystal structures used in this work are available in the RSCB Protein Data Bank (https://www.rcsb.org/) under the codes listed in the paper. The source data for the tables and plots presented in the paper and in the Supplementary Information are also provided along with the paper as the Source Data file. Source data are provided with this paper.

## Code availability

The components of the SQM2.20 scoring function were calculated using MOPAC (http://openmopac.net), AmberTools (https://ambermd.org), and Cuby4 (http://cuby4.molecular.cz), all of which are open source software. The other calculations presented here were performed using the software packages referenced in the Methods section above, which are available from their authors under various licenses.

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

## Acknowledgements

This work was supported by the Institute of Organic Chemistry and Biochemistry of the Czech Academy of Sciences and the computer time was provided by the Ministry of Education, Youth and Sports of the Czech Republic through the e-INFRA CZ (ID:90254). We would also like to thank prof. Pavel Hobza for continuous support.

## Author contributions

AP, JF, ML and JŘ contributed extensively to the conception and design of the current work supervised by JŘ; AP and JF jointly performed most of the calculations and AP, JF, ML and JŘ, analyzed data, prepared and revised the manuscript.

## Competing interests

The authors declare the following competing interests: IOCB Prague, the employer of the authors, is licensing the know-how on the SQM-based scoring function within a collaborative project funded by a U.S.-based pharmaceutical company. This funder had no role in the conceptualization, design, data collection, analysis, decision to publish, nor preparation of the manuscript.
