## [Peer Review File · Nature Communications]

Reviewers' Comments:

Reviewer #1:

Remarks to the Author:

In this manuscript, the authors have developed a new scoring method for relative protein–ligand binding affinities, based on semiempirical PM6-D3H4X/COSMO2 calculations on models including all atoms within 10 Å of the ligand. They have also developed a new test set of 164 accurate binding affinities involving 10 series of ligands binding to different proteins. They show that this SQM2.20 approach gives excellent correlation for all targets ($R^2 = 0.56-0.81$) and strongly outperforms 22 scoring functions, MM, as well as even DFT calculations (slightly). Results can be obtained in minutes for each complex. These are very impressive results and clearly merit publication.

1. The authors should specify what they mean by “lost correlation”.
2. What method was used to calculate the solvation energy at the MM level?
3. It is unclear what is the difference between the DFT results of $R^2 = 0.62$ and 0.64 .
4. It is unclear why DFT failed because of the smaller model used for BACE.
5. Do all R^2 scores in Table 2 have a positive R coefficient? Otherwise, they should be marked with a negative R^2 .
6. How are the residues truncated in the 10 or 6-Å models? Are all atoms included or only functional groups?
7. The “gradual series of optimizations” should be described in detail.
8. How did you know that the ligand changed its protonation state upon binding? What criteria were used?
9. “partially optimized” need to be specified: exact which atoms were optimized (the results should be reproducible).
10. The only quality measure used throughout the article is the correlation coefficient. The authors must add a discussion also of the energies. Since absolute energies are not reliable, they can subtract the average experimental or calculated energy and then calculate the mean absolute deviation and the slope of the best correlation line for the relative energies (for each protein). Methods giving good correlation often give a too high slope. A picture of the calculated vs experimental energies would also be a useful way to illustrate the good results.

Reviewer #2:

Remarks to the Author:

The manuscript describes development of binding energy scoring function based on semi-empirical QM methodology and creation of a small, curated dataset of crystal structures that is used for methodology development.

Scoring of the experimental, or generated ligand-biomolecule interaction is a long standing problem hindering target-based drug discovery. In this context any significant development is of great scientific interest.

However, the presented manuscript does not constitute sufficient progress over the state of the art to justify publication in Nat. Comm.

Below, please see detailed discussion of the manuscript issues:

Overall, the manuscript can benefit from clearer language. I do consider this manuscript to fall on the category of “method development”, as this describes a simplified thermodynamic model for quickly scoring inhibitors in a protein. In this sense, I believe that additional detail regarding method development is necessary. Especially since this is the third manuscript published by these authors on the same methodology, and in each publication there are fine differences. These are addressed in the questions listed below. Additionally, I consider that more details regarding system preparation should be given. In general, I found that the description of the preparation procedures, which can be quite significant and cumbersome, were fully disregarded and left untouched. I understand there might be issues regarding conflict of interests. Still, there should be

a balance between the conflict of interest and transparency, allowing this manuscript to be truly useful for all possible users of SQM2.20.

Questions:

1) On the data quality:

a) What are the resolutions of the crystal structures selected for PL-REX? I did not see this information compiled either on the main text nor on the SI. This should be particularly important for the PDB-ID from which protein was extracted.

b) Did the authors check whether the ligand's electronic density is well resolved for the selected binding poses?

2) At the end of first page of introduction (page 1 in the pdf I got), third paragraph, it is said that a single protein conformation was chosen for each protein set. Ligands from other complexes with the same protein were inserted by overlapping crystal structures. I fully agree that this removes bias from different protein conformations, but it can easily be envisioned that such an approach may bias the calculations towards a specific ligand, or a specific set of ligands. I can easily claim and find arguments stating that this approach biases the results in order to improve the correlation coefficient obtained for SQM2.20. I think it is important not to undersell what was done to ensure this minimal bias, because it will give more value to the work here described. It is said in the manuscript that an SQM/MM setup was used on ligand and its surroundings, without further details (later it is mentioned that PM6-D3H4X is used for ligand, but nothing else).

a) What exactly are the methods used, which cutoff, and which measures were applied to ensure minimal bias? Would the protein preparation wizard yield identical results to the ones reported? This could be for instance used to give information regarding system preparation without disclosing the procedures followed exactly.

b) I believe it would be fair/transparent to compare the scoring for native crystal pose against the prepared ones. It is true that the authors compare in the SI against MM-optimized structures. However, MM-optimized structures are still artificial, even more so than the quantum chemical ones. I know that crystal structures have artefacts, they may have clashes, they lack protons, etc., but this is the only true experimental data available. How is this influencing SQM2.20 scoring?

3) On page 2, it is mentioned that the reparametrized variant of PM6-D3H4X is used for scoring. In the materials and methods at the end it is mentioned that the "native" MOPAC parameters lead to deterioration of the correlations.

a) I would suggest adding the results to SI (and mention it in the main text). Here I mean the correlation plots equivalent to Figure S1 (I suppose they were prepared already, so it should be just inclusion of these).

4)

a) The same for the comparison between solvation models (COSMO2 Vs. COSMO).

5) Both points 3 and 4 would strengthen the use of CUBY as interface. As CUBY is part of the authors' development, I would suggest not underrating this point in text.

6) How exactly is the change of protonation state included in the SF? As far as I understand the methods and materials, this is just the difference in energy between different protonation states of the ligand evaluated in implicit solvent. This is a very crude approximation to proton affinities that biases at many levels. I would even question the necessity to include such term in the scoring function, since it might end up adding more uncertainty and reducing the agreement of the proposed SF and the experimental data.

a) I think that placing a generic chemical equation that explains the acid-base reaction would make interpretation easier and clear in terms of the computational procedure.

b) It would be interesting to compare, when suitable, the impact for including this term. Perhaps the best way is to "recalculate" SQM2.20 without this term and regenerate the plots in Figure S1. Some discussion would make the text richer and most convincing as to why this term should be included in the present form (or not). Note that I am not questioning the need to include proton exchange terms in the scoring function. I am questioning the ability of PM6-D3H4X to recover meaningful proton affinities. Note that Stewart introduced a specific method for proton affinities of oxygen-related acid-base reactions. The necessity for proton-affinity specific parameterization

speaks against direct use of PM6-D3H4X for that purpose.

c) what is the proposed strategy when pKa's are unknown? I understand from the text that this is included whenever the ligand's pKa is known, as it allows "experimental determination" of the contribution of the several acid/base species.

7)

a) Similar considerations apply to other terms in the scoring function.

b) If I am correct, in the first version of the SQM scoring function, the conformational entropy was estimated from a simple penalty over each and every rotatable bond (TDS \sim 1kcal/mol for each rotatable bond). What is the impact of neglecting conformational entropy for scoring?

c) What would happen if the previous approach would have not been replaced? Maybe a full analysis is not warranted, but a few comments should be possible. Please keep in consideration point 20 of this review.

d) Also, the LM5 correlation used for Sconf is formally valid for ligands with up to 200 atoms (if I am not mistaken). Is your variant of the model also limited to similar ligands? Did you parametrize it equivalently for use together with PM6-D3H4X? If LM5 was reparametrized, more details should be given in SI.

8) MOZYME can be employed with even larger systems. According to the MOPAC webpage there is a cutoff at 15k atoms. Surely, a scoring function needs a certain degree of efficiency for quick evaluation, therefore justifying the limit to 2k atoms (this should be explicit in the text, the reason why). However, cutting a protein leads to overhead too, or at least it leads to additional considerations that may render the analysis less straightforward. Regions that are cut need some sort of capping (which one is used?), charge distributions are affected, All of this must be properly considered. Though the protein preparation wizard of Schrodinger makes life easier in many aspects, that requires having a license, which is not affordable for many researchers. Of course, there are alternatives, but they all lead to additional considerations, no matter how minimal those are.

a) I would consider detailing the procedure used necessary for reproducibility of the results. Again, I understand that this gets close to the conflict of interest declared, so a compromise protecting your interests.

b) This is only a suggestion, so it remains for the authors to consider: I would suggest estimating the error from this atom cutoff, i.e., comparing the impact of 2k atom cutoff against that of full protein (or 15k atom cutoff). Note that this question comes naturally (at least to me) when I read the third paragraph of the section SQM2.20 scoring (last of page 3). Then it occurred to me that the ligand set contains charged molecules, and electrostatics are long-range interactions...

9) In the last paragraph of the introduction it is mentioned that SQM2.20 is meant to fill the gap between very fast but inaccurate SFs and high-level DFT calculations where additional system constraints are needed. However, I see a very shy mentioning of timing information regarding SQM2.20 in the text. I believe such information should be duly pointed in the main text, already in the introduction, as indicated at the beginning of this point.

a) I would suggest adding a plot with exact calculation times for PM6-D3H4X and systems with \sim 2k atoms. The plots I would make are according to protein, showing the calculation times for each complex. This is just a suggestion though. Alternatively, consider adding more statistical descriptors (max, min, ...). Note that you can add this data to the SI with respective mentioning where it can be found. If you just say in the main text that the average calculation time is 10 minutes, that is great and more precise than "a few minutes". I know some people would criticize saying it takes too long. But you show the significant gain in accuracy offered by your SF. And like you mention, SQM2.20 is supposed to fill in the gap between high-level and low-level. Therefore, it is supposed to be applied to 100-1000 structures. 5 minutes waiting time is perfectly fine. (5 minutes is just example).

b) From a psychological point of view, the 643 hours required by DFT on average is a lot. I don't need to think any further to be scared and immediately demotivated. But 0.05 hours mean nothing, and I am "emotionless" towards SQM2.20's efficiency. I would suggest putting in parentheses the times in minutes. Just a psychological thing.

10) It is also mentioned at the end of introduction that SQM2.20 may be used at any stage of CADD. However, as far as I understood, there is no interface provided. So, the users would have to

implement their own workflow? Is it offered as a service? I would recommend clarifying this point, making special reference if and when the authors plan to provide an interface. If something will be available from CUBY (to protect conflict of interests), it would be interesting to know. It is also OK to simply mention that implementation and adaptation of the workflow to each user's environment is straightforward.

11) A medicinal chemist that might be interested in using SQM2.20 would ask for the structures of the ligands for each protein. A 2D representation like that provided by Chem Draw or Marvin Sketch (or anything equivalent) would be nice. If possible, choosing a representation that reflects the orientation of the ligand inside the binding pocket for each set, or a representation faithful enough of the relative binding poses.

12) PL-REX contains metalloproteins with Zinc. If I am not mistaken, in previous work of the authors, it is mentioned that PM6-D3H4X has difficulties in treating the Zinc atoms. It is also mentioned that DFTB3, parametrization 3OB, offers a much more robust evaluation of scoring when Zinc is present. Also, in this publication (10.1021/acs.jcim.9b01171), you show how great DFTB3 is. You invested in the development of the H5 correction for DFTB3, which is newer than D3H4X. This is particularly interesting to analyse further.

a) Since these publications (present and 10.1021/acs.jcim.9b01171) discuss similar topics, maybe a few comments regarding how the overall results compare are possible?

b) I am curious to read a comment from the authors on PM6 Vs. DFTB. Did the authors abandon DFTB3 as alternative for scoring? Note that this does not necessarily imply any change in the main text because DFTB is obviously not the subject and is completely disregarded.

c) Did the authors try to use SQM2.20 on the datasets offered in 10.1021/acs.jcim.9b01171. Is the correlation still as good? This does not have to involve any change in text or I don't expect any additional calculations regarding this subject.

13) The plots in SI, section S2 show the comparison between SQM2.20 score and the experimental Gibbs free energies of binding, for all targets. As far as the description goes in SI, not all systems have K_i documented. Some seem to have exclusively IC_{50} . Though IC_{50} is proportional to K_i , the proportionality constant is not always 1. Further, K_i is an equilibrium constant in the presence of a substrate. In my opinion, a true thermodynamic comparison between the SQM2.20 scoring, and the K_i requires including the substrate, which was never mentioned anywhere in the text (I assume it was fully neglected). In my interpretation, DG_{bind} refers to the reaction $P + I = PI$, meaning that it relates to K_d . Surely K_i also reflects binding, but the impact of the natural substrate should be made explicit in the thermodynamic expressions.

a) In this sense I believe it is thermodynamically incorrect to call the experimental affinities "experimental DG_{bind} " in Figure S1. In particular, the substrate will also have binding, entropy, proton-exchange contributions that will affect binding of the inhibitor. Furthermore, IC_{50} is to be interpreted as a concentration, not an equilibrium constant. pK_i or pIC_{50} are good enough.

14)

a) What is the criterion for "confidence" used for generating PL structures for ligands with no crystal structure?

15)

a) Schrodinger's protein preparation wizard determines the protonation states of the ligand in the pocket. How do these compare to the ones proposed by the authors?

16)

a) Though clear for me, I think it should be stated in page 9 why PM7 fails in the scoring. This will help unexperienced users choose their quantum chemical method.

17)

a) A diagram with workflow would make everything clearer from the protocol point of view.

18)

a) I would show plots in Figure S1 for other SFs. I suppose these were previously prepared, so they should simply be included for transparency.

19)

a) I suppose that calculations are ran on charged systems, i.e., no charge neutralization is used. This should be explicit.

20) The work of Chan and coworkers used for the quick estimation of entropy effects clearly shows that vibrational contributions to the GFN2-xTB Svib are not negligible. Further, neglecting these should have stronger impact on the thermodynamics of binding than Sconfig. I would be interested in comments/thoughts on this. Note that this does not necessarily require changes in the manuscript.

21) At the end of the results/discussion, it is mentioned that "Furthermore, as a non-empirical SF consisting of physically well-defined terms, it provides insight into the nature of P-L interactions, which may guide the rational design of better ligands."

a) There is no discussion regarding this subject in this manuscript, as to how the nature of PL interactions as described by the SQM calculations could guide rational ligand design. I would even claim that if I were to take 2 ligands and put them inside the protein, there is nothing in SQM2.20 telling me why one ligand is better than the other: the introduction of a fluorine leads to better dispersion, or electrostatics? There is only one singular value, the scoring, that predicts whether binding is improved. There is no rationale behind it, except potential user interpretation which, for inexperienced users may be biased by wishful thinking. I would either suggest removing this sentence, as it is misleading with regard to SQM2.20's scope or make reference to quantum chemical approaches that can cover such guidance scenarios.

22)

a) GlideSP offers better results than GlideXP. Maybe a sentence to comment on this? Even if it is just saying that you noted this, but there is no clear reason as to why since you don't have the source code nor developed any of the methods. Anyhow, it could bring awareness regarding the need for more trustworthy scoring functions, like the one you are proposing.

Dear Reviewers,

First of all, we appreciate the effort you put into reviewing our manuscript and thank you for the detailed and constructive comments that helped us improve the manuscript. Below, please find detailed point-by-point responses (blue text) to all the raised questions of the reviewers (black text), along with the description of the changes we have made to the manuscript and the Supporting Information. The removed text is indicated by strikethrough (~~removed~~) and the added text by underlining it (added). We also provide a manuscript file with highlighted changes (blue text). We believe that these changes have helped improve the scientific quality of the paper and the clarity of its presentation.

Reviewer #1

In this manuscript, the authors have developed a new scoring method for relative protein–ligand binding affinities, based on semiempirical PM6-D3H4X/COSMO2 calculations on models including all atoms within 10 Å of the ligand. They have also developed a new test set of 164 accurate binding affinities involving 10 series of ligands binding to different proteins. They show that this SQM2.20 approach gives excellent correlation for all targets ($R^2 = 0.56$ – 0.81) and strongly outperforms 22 scoring functions, MM, as well as even DFT calculations (slightly). Results can be obtained in minutes for each complex. These are very impressive results and clearly merit publication.

1. The authors should specify what they mean by “lost correlation”.

We agree that this is a vague formulation, we rewrote the sentence on pg. 4 using the actual number:

“First, SQM2.20 scores calculated on MM-optimized geometries resulted in a significant drop of correlation ($R^2 < 0.36$) ~~lost correlation~~ in three targets, while ~~and~~ the average R^2 dropped to 0.52 (Table 2)”

2. What method was used to calculate the solvation energy at the MM level?

We used the Generalized Born model IGB7 to calculate the solvation implicitly at the MM level. We describe this on pg. 4 in the following sentence, and we cite the appropriate reference:

“The effect of using SQM calculations was assessed by stepwise replacing the geometries and energy terms of the scoring protocol with their MM equivalents (using AMBER ff19SB/GAFF2 force fields^{22,23} with IGB7 implicit solvent²⁴).”

We describe it in more details in the “AMBER scoring” Section in Methods on pg. 14:

“AMBER scoring refers to the MM analogue of SQM2.20 where the ΔE_{int} , $\Delta \Delta G_{\text{solv}}$ and $\Delta G_{\text{conf}}(L)$ terms of Eq. 1 are evaluated at the MM level. We used the AMBER ff19SB force field²² for the protein and GAFF2 force field²³ with partial charges extracted from

PM6 calculations for the ligands; the environment was described by the IGB7 implicit solvent model.²⁴

3. It is unclear what is the difference between the DFT results of $R^2 = 0.62$ and 0.64 .

In the paper, we compared SQM2.20 results with DFT scoring on the trimmed models of the PL-REX dataset. First, we discuss the effect of the truncation of the model, still at the SQM level. Here, the complete PL-REX dataset is used, and the average R^2 is 0.62 .

However, the DFT was not applicable to the 04-AR target, and we also exclude 06-BACE from the comparison because it is clear that the smaller model does not work (for details see our response to point 4). Once these are excluded, SQM2.20 yielded an average R^2 of 0.67 and DFT has R^2 of 0.64 . The results on the incomplete dataset are indicated with a footnote "a" in Table 2.

However, we admit that the original description may have been confusing and thus we have rewritten the following paragraph on pg. 5 to make it clearer.

"The results obtained with these trimmed models using SQM2.20, with the exception of 06-BACE1 ($R^2 = 0.37$), still correlated well with experimental data (average $R^2 = 0.62$, Table 2) but at the level of individual targets, the correlation deteriorated significantly in 06-BACE1 ($R^2 = 0.37$). In this target, the truncation of the active site model results in an unphysically large molecular charge (+5), which is only compensated by the inclusion of more distant anionic amino acid residues in the larger model. This shows the importance of including larger protein surroundings of their ligands for consistently excellent performance, as is the case with the default model. The DFT scoring was evaluated in the eight systems (excluding 06-BACE for structural reasons already demonstrated at the SQM level and 04-AR where DFT failed to converge in multiple iodine-containing ligands). In this subset set of eight targets, SQM2.20 with the trimmed models yields an average R^2 of 0.67 , and the DFT scoring with the trimmed models yielded a very similar result with an average R^2 of 0.64 (Table 2). in the eight systems where DFT was applicable (analogously to SQM scoring, 06-BACE failed because of the smaller model used; we had to exclude also the 04-AR target, where DFT calculations of iodine-containing ligands failed to converge)."

4. It is unclear why DFT failed because of the smaller model used for BACE.

This case is not a failure of DFT, but a failure of the smaller model itself - as already discussed above, both SQM2.20 and DFT failed there. This has a structural reason: there is a large number of cationic amino acids in a surrounding layer of about 5 \AA around the ligands, which results in a total charge of +5 in the smaller model. This positive electrostatic potential is compensated by a number of negatively charged amino acids in the next layer of about $5 - 9 \text{ \AA}$ from the binding site. Consequently, the larger receptor model of 06-BACE in PL-REX, for which SQM2.20 scoring works, has a total charge of -2. Moreover, the overall charge of the smaller model, +5, is the largest value of all those investigated.

We have added this explanation to the paper on pg. 5 where this failure is discussed (Results and discussion/SQM2.20 scoring Section):

“In this target, the truncation of the active site model results in an unphysically large molecular charge (+5), which is only compensated by the inclusion of more distant anionic amino acid residues in the larger model.”

5. Do all R2 scores in Table 2 have a positive R coefficient? Otherwise, they should be marked with a negative R2.

Most of the correlations reported in Table 2 do have positive correlation coefficients with the exception of AMBER SF in the 09-CDK2 and 10-MMP12 targets. We newly indicate these as “correlations with a negative value of R” with a footnote “c” in Table 2.

6. How are the residues truncated in the 10 or 6-Å models? Are all atoms included or only functional groups?

We include the entire residues where at least one atom falls within the cutoff distance, and we add caps that properly close the peptide bond of the truncated backbone. We have added this information to the manuscript, the relevant paragraph in the Methods Section now reads (with the additions underlined):

“For each target, the default models were defined as residues of the representative protein within 10 Å of all the overlaid ligands and amounted to 1,295–2,298 atoms (a residue is selected if at least one of its atoms fits within the cutoff distance). The trimmed models comprised 6 Å surroundings (only in the cases of 02-HIV-PR and 06-BACE it was reduced to 5 Å because of the large size of the ligands) of all the overlaid ligands and were made up of 753–1,096 atoms. The protein backbone was then terminated with neutral caps that close the peptide bond and introduce the least perturbation into the system (-NH-CH₃ at the C-terminus and -CH=O at the N-terminus).”

7. The “gradual series of optimizations” should be described in detail.

We have clarified the optimization protocol used to prepare the structures, extending the original brief note:

“The P–L models were gradually relaxed in a series of optimizations. The final step was a free gradient optimization of the ligand and its surroundings ...”

to:

“The P–L models were gradually relaxed in a series of optimizations. First, we performed annealing (60 ps molecular dynamics with a thermostat cooling the system from 300 to 0 K) and optimization of the hydrogen atoms in the protein only. Second, we formed the P-L complexes, and possible clashes with the protein were resolved by local geometry optimizations at the MM level. The final step was a free gradient optimization of the ligand and its surroundings ...”

Additionally, we clarified the setup of the final SQM/MM optimization, as we describe in the response to point 9), and we now explicitly note that the SQM/MM implementation is provided by Cuby:

“The optimizations were performed in Cuby4 (<http://cuby4.molecular.cz>),²⁷ which also provided the interface that implements the SQM/MM scheme.”

It has to be noted that the first step of the preparation, the molecular dynamics, is not deterministic, and also the ad hoc optimizations resolving the clashes would be hard to reproduce exactly. This is why we provide the final structures we used for the scoring in the PL-REX repository. These geometries can be used as a starting point for reproducing the SQM2.20 scores reported in the manuscript, and in the scoring protocol itself there are no such ambiguities. On the other hand, we have verified that the random factor introduced by the MD step does not affect the final results significantly, and that the published results represent well the mean of the distribution obtained by repeating the simulations multiple times.

8. How did you know that the ligand changed its protonation state upon binding? What criteria were used?

For the scoring, the protonation state of the ligand is determined in the protein-ligand complex. The description of this approach has been improved in the revised manuscript and is described in more detail below in our response to point 15 of Reviewer 2. Automated tools for estimating protonation are not reliable, we visually inspected all systems and manually corrected the protonation state where necessary. If this protonation does not match that of a free ligand (determined from the pKa of the ligand and pH used in the experiment), the protonation penalty is added to account for the difference between these two states. The description of this approach has also been expanded in the "Methods/Scoring/SQM2.20 SF" Section.

9. “partially optimized” need to be specified: exact which atoms were optimized (the results should be reproducible).

We agree that this is important information and we add it to the manuscript. All protein residues within 4 Å of all the ligands are selected, analogously to the selection of the model system itself. We added it to the description of the optimization of the PL-REX dataset structures (Section “Methods/PL-REX dataset”):

“The final step was a free gradient optimization of the ligand and its surroundings (all protein residues within 4 Å of all the ligands) ...”

and we refer to this in the description of AMBER-only scoring where this partial optimization is mentioned again:

“AMBER SF was used for the scoring of P–L complexes which were partially optimized (ligands and their close surroundings defined in the same way as in the optimization of the PL-REX structures) at the MM level”

10. The only quality measure used throughout the article is the correlation coefficient. The authors must add a discussion also of the energies. Since absolute energies are not reliable, they can subtract the average experimental or calculated energy and then calculate the mean absolute deviation and the slope of the best correlation line for the relative energies (for each

protein). Methods giving good correlation often give a too high slope. A picture of the calculated vs experimental energies would also be a useful way to illustrate the good results.

We are aware of the fact that the scale of the score does not correspond to the absolute binding free energies, and that the slope of the linear regression of the data is higher than 1. This is inevitable for a scoring function that calculates only some of the terms of the actual free energy and does not apply any empirical scaling. The main cause of this problem is the neglect of the thermal effects and the dynamics of the system, which would weaken the interaction compared to our calculation on a single optimized structure.

To clarify this in the paper, we are extending the short note on this (which was already present) in the “Scoring” subSection of “Methods”, with a reference to a more detailed discussion that we have added to the Supplementary Information. It now reads (with the new text underlined):

“Neither SQM2.20 nor most of the other scoring functions tested yield absolute binding free energies for which error values (in comparison to experimental data) can be expressed in energy units (the relationship between the experimental ΔG_{bind}^0 and the SQM2.20 score is discussed in more detail in the SI, Section S8).”

In the Supplementary Information, we are adding a new Section “S8) Relationship between SQM2.20 and absolute binding free energy”, with additional discussion, table of the slopes in the individual series of ligands and a plot of the scores as suggested by the reviewer.

Reviewer #2

The manuscript describes development of binding energy scoring function based on semi-empirical QM methodology and creation of a small, curated dataset of crystal structures that is used for methodology development.

Scoring of the experimental, or generated ligand-biomolecule interaction is a long standing problem hindering target-based drug discovery. In this context any significant development is of great scientific interest.

However, the presented manuscript does not constitute sufficient progress over the state of the art to justify publication in Nat. Comm.

Below, please see detailed discussion of the manuscript issues:

Overall, the manuscript can benefit from clearer language. I do consider this manuscript to fall on the category of “method development”, as this describes a simplified thermodynamic model for quickly scoring inhibitors in a protein. In this sense, I believe that additional detail regarding method development is necessary. Especially since this is the third manuscript published by these authors on the same methodology, and in each publication there are fine differences. These are addressed in the questions listed below. Additionally, I consider that more details regarding system preparation should be given. In general, I found that the description of the

preparation procedures, which can be quite significant and cumbersome, were fully disregarded and left untouched. I understand there might be issues regarding conflict of interests. Still, there should be a balance between the conflict of interest and transparency, allowing this manuscript to be truly useful for all possible users of SQM2.20.

The reviewer is correct in pointing out the weaknesses in describing the novelty of the SQM2.20 method and its difference from our prior work. We revised the manuscript to address that.

Firstly, we have expanded the description of our methodology by including the preparation procedures used and we have also made several changes that highlight the differences between our previous work and the current study. The overall approach is, of course, similar because we are trying to calculate the actual physical components of the binding free energy using similar quantum mechanics-based methods. However, virtually every component of the current scoring function (SF) has been updated with the latest methodological developments which, as a whole, have significantly improved the quality of scoring and made the SF more robust. We have also added a new term describing possible changes in the protonation states of the ligands upon binding to their target proteins, which increases the generality of the SF (discussion of this term has also been expanded in the revised manuscript). Another major advance is that we have streamlined the current SF to use the most computationally efficient methods applicable to each sub-task. In contrast to some of our previous studies which used more demanding computations, the current SF is truly as fast as possible at the SQM level. These developments have been summarized in the Introduction, where we have extended the relevant part which now reads (new text underlined):

“Over the last decade, we have been developing SFs based on SQM calculations and applied them successfully to various targets and tasks,^{7,8} which had been reviewed in refs. 7 and 8. In our previous studies, we used different methods, often tailored to a specific problem or even to specific targets. In some cases, we opted for computationally more demanding quantum-mechanical methods that can not be used on a larger scale. Here, we leverage our experience to formulate a universal SQM-based SF, named SQM2.20, for general use across diverse protein targets, various ligand chemistries and modes of non-covalent interactions. It is free of any empiricism (apart from system-independent parameters in the underlying computational methods) and is neither tuned to a specific target, nor to protein–ligand interactions in general. The SQM2.20 SF covers the most important contributions to P–L binding free energy, and all these terms have been updated to using the best methods available at this computational level. Together, these changes led to a significant improvement over our previous work. Moreover, this was achieved without compromising the excellent computational efficiency.”

More details on how the individual changes in the methodology improve the results are now discussed better in the text, including explicit comparison to the results obtained with some of the methods we used previously. This is described e.g. in our response to the points 6), 7) and 12) below.

Secondly, we have also improved the description of the procedure used to prepare the systems for calculations, as suggested. However, we must note that the preparation requires

actions which cannot be automated, and we thus provide all the structures we prepared as a starting point for reproducing the scoring itself (this is also mentioned in the manuscript). Here, we would like to point out that such a collection of carefully curated, highly accurate experimental data prepared for computation, provided along with the final optimized structures, is extremely important for the development and evaluation of any method applicable to protein-ligand interactions. The PL-REX dataset made freely available to the scientific community thus represents another key advance of the current work.

Below, we address all the specific points.

Questions:

1) On the data quality:

a) What are the resolutions of the crystal structures selected for PL-REX? I did not see this information compiled either on the main text nor on the SI. This should be particularly important for the PDB-ID from which protein was extracted.

We added crystallographic resolution for the protein-ligand complexes used as the representative structures for scoring to Table 1 in the main text. For all the PDBs used for extracting the geometry of the ligand only, we added this information into Table S7 in the Supplementary Information.

b) Did the authors check whether the ligand's electronic density is well resolved for the selected binding poses?

As an objective measure of the “goodness of fit” of the ligand structure modeled by the crystallographers into crystallographic electron densities, we use Real Space Correlation Coefficient for the ligands. We added this information into Table S7 in the Supplementary Information. In the vast majority of the structures, the ligands are resolved with very high confidence (average RSCC is 0.94 on the scale from 0 to 1). In the few structures with lower (but still rather good) RSCC (the minimum is 0.75) or in cases where this metric was missing, we visually inspected the electron densities in the Coot program, version 0.9.8, and verified that the binding pose is correctly and unambiguously determined. To reference the new data in the SI from the main text, we added a following sentence to the description of the PL-REX dataset at the very beginning of Methods Section:

“For crystal structure quality metrics, see the SI, Section S9.”

2) At the end of first page of introduction (page 1 in the pdf I got), third paragraph, it is said that a single protein conformation was chosen for each protein set. Ligands from other complexes with the same protein were inserted by overlapping crystal structures. I fully agree that this removes bias from different protein conformations, but it can easily be envisioned that such an approach may bias the calculations towards a specific ligand, or a specific set of ligands. I can easily claim and find arguments stating that this approach biases the results in order to improve the correlation coefficient obtained for SQM2.20. I think it is important not to undersell what was done to ensure this minimal bias, because it will give more value to the work described here. It is said in the manuscript that an SQM/MM setup was used on ligand

and its surroundings, without further details (later it is mentioned that PM6-D3H4X is used for ligand, but nothing else).

a) What exactly are the methods used, which cutoff, and which measures were applied to ensure minimal bias? Would the protein preparation wizard yield identical results to the ones reported? This could be for instance used to give information regarding system preparation without disclosing the procedures followed exactly.

We agree that it is important to explain more carefully how the protein conformation was selected and how it influenced the result. With the exception of one target where only few crystals were available, the representative structure was selected systematically using the same, rational rules. This selection of course favors good results, but for a good, structural reason. And because we use the same rules for all the systems, this can not be considered as a deliberate bias. For the remaining one target, we chose the structure which was the best starting point for modeling the systems without experimental structure.

First, we clarify how the crystal structures were selected in the Method Section of the paper. We have replaced the paragraph describing the protein selection in the Methods/PL-REX dataset Section:

“For most targets, we chose the protein structure having the least close contacts with all the aligned ligands (protein crystals with missing side chains in the binding sites were discarded prior to the selection). Only for the 05-Cath-D target, we chose the structure with a ligand which was the most similar to the compounds modeled manually.”

with more elaborate description, also referring to the newly added information in the SI:

“For 9 out of 10 targets, we systematically chose the protein structure that could best spatially accommodate all the aligned ligands. The closest atom-atom distance between each protein structure and all the ligands was measured as an indication of the ability to accommodate the ligands, and the protein with the largest measured value was selected. Protein crystals with missing side chains in the binding sites were discarded prior to the selection. Only in the case of the 05-Cath-D target, where only 3 crystal structures are available and 7 ligands were built, we selected the native protein structure of a ligand that was the most similar to the manually modeled compounds. To evaluate how the choice of the receptor structure affects the results, SQM2.20 was also tested on protein structures selected according to different criteria. In addition, in the case of the 01-CA2 target, we also scored each ligand in its own native crystal structure. Both these analyses are discussed in the Supplementary Information, Section S7).”

Second, we demonstrate the sensitivity of the results to the choice of the reference crystal structure by performing additional calculations with proteins selected using different criteria, and we analyze the cases where this led to worse scoring results. This was added to the SI, Section “S7” and Table S5, and is referenced in the updated text above.

It should also be mentioned that in order to avoid any bias, we use all the ligands (for which structure and affinity are reliably known) from the original sources, and we do not discard even the problematic ligands which lead to worse correlation. We add a note on that to the SI Section S1 where all the targets are described and the sources are cited, and we reference it in the main text at page 4 by adding the underlined note:

“... other complexes were built by modifying ligands closely similar to those for which crystal structures were available (following the rules listed in the SI, Section S1).”

Also, to highlight this in the main text, we add the following sentence to the paper where the PL-REX dataset is described in the Introduction:

“No ligands from the original sources that met the above criteria were arbitrarily discarded, even if they were difficult to score and negatively affected the final results.”

b) I believe it would be fair/transparent to compare the scoring for native crystal pose against the prepared ones. It is true that the authors compare in the SI against MM-optimized structures. However, MM-optimized structures are still artificial, even more so than the quantum chemical ones. I know that crystal structures have artefacts, they may have clashes, they lack protons, etc., but this is the only true experimental data available. How is this influencing SQM2.20 scoring?

We agree that it is interesting to see how this affects the SQM2.20 scoring. However, preparing the native crystal for each ligand is very demanding, and our prior experience shows that scoring in multiple crystals does not lead to good results. To demonstrate this in the paper, we are adding such an analysis performed on one example, the 01-CA2 target, where all the ligands have a well-resolved crystal. There, we scored each ligand in its native crystal structure, and we added the results and discussion to the SI Section S7. In particular, the following paragraph has been added:

“Finally, each ligand can be scored in its own crystal structure. We do not use this approach because the scoring function neglects some terms that may cancel if a single protein structure is used for all the ligands – this is already discussed in the paper. Also, preparing multiple crystals for the calculations and ensuring that they are as consistent as possible is a tedious task. As a demonstration, we performed such an analysis for the 01-CA2 target where each ligand was scored in its own crystal. The computed scores were slightly more favorable than when using a single receptor structure, on average by 0.7 kcal/mol. However, the obtained correlation R^2 of 0.43 was significantly lower than that when using a single receptor structure where R^2 ranged from 0.52 to 0.67 (Table S5). This result highlights the importance of using a single representative protein structure when an end-point scoring function of this type is used.”

On pg. 11 of the manuscript, we also add a note on this with a reference to the SI:

“... In addition, in the case of the 01-CA2 target, we also scored each ligand in its own native crystal structure. Both these analyses are discussed in the SI, Section S7).”

3) On page 2, it is mentioned that the reparametrized variant of PM6-D3H4X is used for scoring. In the materials and methods at the end it is mentioned that the "native" MOPAC parameters lead to deterioration of the correlations.

a) I would suggest adding the results to SI (and mention it in the main text). Here I mean the correlation plots equivalent to Figure S1 (I suppose they were prepared already, so it should be just inclusion of these).

See the response to 4a) below.

4)

a) The same for the comparison between solvation models (COSMO2 Vs. COSMO).

To the points 3) and 4), we added plots analogous to Figure S1 to the Supplementary Information as suggested, for both the variants of the SF using default PM6-D3H4X/COSMO and PM7/COSMO. More details on this new part of the SI are provided below in the response to point 18). We added a reference to these plots in the paper at the points where these SFs are discussed:

“The individual results of the SF variants based on the default PM6-D3H4X/COSMO and on PM7/COSMO calculations are also plotted in the Part 2 of the SI.”

5) Both points 3 and 4 would strengthen the use of CUBY as interface. As CUBY is part of the authors' development, I would suggest not underrating this point in text.

Thanks for the supportive comment. Actually, providing a few more details following this suggestion helps to clarify how the calculations are implemented, which should be useful to the readers. We added a note on using Cuby to provide the corrections and to simplify the COSMO2 calculations in the Methods/Scoring/SQM2.20 SF Section (new text underlined):

“The latest parameterizations of the D3H4X corrections were utilized,¹⁵⁻¹⁷ which is simplified by interfacing the customizable implementation of the corrections in Cuby4 to an unmodified version of MOPAC. Similarly, the Cuby4 interface to MOPAC automates the setup and evaluation of the COSMO2 solvation free energy.”

We also added a note that Cuby is used to perform the SQM/MM calculations, which is described in our response to the point 7) of reviewer 1.

6) How exactly is the change of protonation state included in the SF? As far as I understand the methods and materials, this is just the difference in energy between different protonation states of the ligand evaluated in implicit solvent. This is a very crude approximation to proton affinities that biases at many levels. I would even question the necessity to include such term in the scoring function, since it might end up adding more uncertainty and reducing the agreement of the proposed SF and the experimental data.

a) I think that placing a generic chemical equation that explains the acid-base reaction would make interpretation easier and clear in terms of the computational procedure.

We agree that this part of the calculation should be clarified. The ΔG_{H^+} term is not only a difference between different protonation states of the ligand evaluated in the implicit solvent,

it also includes the corresponding energy difference between the H₂O and H₃O⁺ or OH⁻ at the other side of the equation, depending on whether the proton is released from a ligand or transferred to a buffer upon ligand binding. The proton transfer energy is scaled by a fraction of the proton transferred estimated from the pH at which the experimental affinities were measured and from the pKa of the titratable group of the ligand using the rearranged Henderson–Hasselbalch equation. The acid-base equations and a detailed description of the ΔG_{H+} term have been added in the “Methods/Scoring/SQM2.20 SF” Section, specifically:

“Depending on whether the ligand (L) was deprotonated (Eq. 2, 4) or protonated (Eq. 3, 5) in solution, the following acid-base reactions were considered

and, consequently, the ΔG_{H+} term is evaluated as

$$\Delta G_{H^+} = \frac{10^{(pH-pKa)}}{1+10^{(pH-pKa)}} \cdot (G(LH^+) - G(L) + G(OH^-) - G(H_2O)) \quad (\text{Eq. 4})$$

$$\Delta G_{H^+} = 1 - \frac{10^{(pH-pKa)}}{1+10^{(pH-pKa)}} \cdot (G(L^-) - G(LH) + G(H_3O^+) - G(H_2O)) \quad (\text{Eq. 5})$$

G(L), G(LH), G(LH⁺) and G(L⁻) are computed at the PM6-D3H4X/COSMO2 level (here we use the label G for a quantity that is the sum of the PM6-D3H4X enthalpy of formation in the gas phase and the COSMO2 free energy of solvation). Since the solvation free energy of OH⁻ and H₃O⁺ is difficult to calculate reliably with SQM methods, the G(H₃O⁺), G(OH⁻) and G(H₂O) are computed as the sum of the PM6-D3H4 gas phase enthalpy and the experimental solvation free energy.³²”

b) It would be interesting to compare, when suitable, the impact for including this term. Perhaps the best way is to “recalculate” SQM2.20 without this term and regenerate the plots in Figure S1. Some discussion would make the text richer and most convincing as to why this term should be included in the present form (or not). Note that I am not questioning the need to include proton exchange terms in the scoring function. I am questioning the ability of PM6-D3H4X to recover meaningful proton affinities. Note that Stewart introduced a specific method for proton affinities of oxygen-related acid-base reactions. The necessity for proton-affinity specific parameterization speaks against direct use of PM6-D3H4X for that purpose.

The ΔG_{H+} term is applied in four targets, namely 02-HIV-PR, 03-CK2, 08-Trypsin and 10-MMP12. There, it is of course possible to evaluate the score with and without this term. The difference was negligible in two of them (02-HIV-PR, 03-CK2), but omitting the ΔG_{H+} term leads to significantly worse results in 08-Trypsin and 10-MMP12 (R² decreased by 0.11 and 0.13, respectively). This was already mentioned briefly in the SI, we are moving this note to the main text of the manuscript and expanding it as follows.

We are adding the results obtained without the protonation penalty and their discussion to the description of the SQM2.20 SF in the Methods Section of the manuscript (page 13) where the ΔG_{H^+} term is introduced. Specifically, we added the following paragraph:

“Further, the effect of omitting the ΔG_{H^+} term from SQM2.20 was assessed. The correlation deteriorated moderately in the 08-Trypsin and 10-MMP12 targets (R^2 decreased by 0.11 and 0.13, respectively) or negligibly in the 02-HIV-PR and 03-CK2 targets (R^2 decreased by 0.03 and 0.06, respectively).”

We are aware of the difficulties with computing proton affinities at the semiempirical level. The results are better for larger organic molecules such as the ligands, the limiting factor being the calculation of the solvation free energies of water ions H_3O^+ and OH^- . To avoid this issue, we use experimental solvation energies for these species. This information has been added to the manuscript as a part of the update described in our response to the previous point.

c) what is the proposed strategy when pKa's are unknown? I understand from the text that this is included whenever the ligand's pKa is known, as it allows “experimental determination” of the contribution of the several acid/base species.

The experimental pKa was only known for the case of the 08-Trypsin ligands, and for 10-MMP12, experimental values of fragments were used. For the other ligands, the pKa values were computed using Stardrop v. 7.0 as it is described in the Methods/Scoring/SQM2.20 SF Section. It should also be noted that the exact value of the pKa becomes important only in a few cases where its value is close to the pH of the buffer.

7)

a) Similar considerations apply to other terms in the scoring function.

The ΔE_{int} and $\Delta \Delta G_{solv}$ are indispensable terms, and together they provide rather good correlation with the experiment even without other contributions. The effect of including the ligand deformation energy is not very strong on the average, but it is important in some of the systems - when it is neglected, the average correlation in the whole dataset remains high with R^2 of 0.66, but the worst result drops to R^2 of 0.38 in the case of BACE1. This term is thus important for obtaining more balanced results. The entropic penalty is discussed in the next point.

b) If I am correct, in the first version of the SQM scoring function, the conformational entropy was estimated from a simple penalty over each and every rotatable bond (TDS ~1kcal/mol for each rotatable bond). What is the impact of neglecting conformational entropy for scoring?

We know that the conformational entropy term as we compute it is not significant for the correlation in the series of ligands investigated there - the results with and without it are practically the same (the average R^2 differs by 0.02, which is negligible compared to the accuracy of the method). However, we do use it in our SF to keep it general - there may be other systems where conformational entropy plays a more important role than in this dataset, and the LM5 model seems to be the best approach available at this computational cost.

c) What would happen if the previous approach would have not been replaced? Maybe a full analysis is not warranted, but a few comments should be possible. Please keep in consideration point 20 of this review.

Using the 1 kcal/mol penalty for each rotatable bond is inferior to the LM5 model; in the PL-REX dataset, a score based on this penalty yields an average R^2 of 0.63.

d) Also, the LM5 correlation used for Sconf is formally valid for ligands with up to 200 atoms (if I am not mistaken). Is your variant of the model also limited to similar ligands? Did you parametrize it equivalently for use together with PM6-D3H4X? If LM5 was reparametrized, more details should be given in SI.

We use the model as it is, and our use safely falls within the limit of 200 atoms.

We are adding a note clarifying the points a) to d) in the Methods Section where the individual terms of the SF are discussed, with a new paragraph, also referencing the simpler entropic penalty model we used previously:

“The conformational entropy term $-T\Delta S$ is computed using the empirical LM5 model,²⁰ which estimates the entropy from several descriptors characterizing the ligand and has been fitted to a large database of GFN2-xTB calculations on drug-like molecules. We use the model as implemented and parameterized by its authors. In the present dataset, the omitting the $-T\Delta S$ term leads to practically the same overall correlation of the score with the experiment. However, when we apply a simpler model which we used previously⁸ (a penalty of 1 kcal/mol per rotatable bond), the results deteriorate slightly to average R^2 of 0.63.”

8) MOZYME can be employed with even larger systems. According to the MOPAC webpage there is a cutoff at 15k atoms. Surely, a scoring function needs a certain degree of efficiency for quick evaluation, therefore justifying the limit to 2k atoms (this should be explicit in the text, the reason why). However, cutting a protein leads to overhead too, or at least it leads to additional considerations that may render the analysis less straightforward. Regions that are cut need some sort of capping (which one is used?), charge distributions are affected, All of this must be properly considered. Though the protein preparation wizard of Schrodinger makes life easier in many aspects, that requires having a license, which is not affordable for many researchers. Of course, there are alternatives, but they all lead to additional considerations, no matter how minimal those are.

The calculations in the whole proteins are possible with MOZYME, we discuss this in the response to point b) below.

a) I would consider detailing the procedure used necessary for reproducibility of the results. Again, I understand that this gets close to the conflict of interest declared, so a compromise protecting your interests.

We have added the information on how exactly the smaller active site model is selected and capped, this is no secret. The details and the additions we made to the manuscript are provided above in our response to point 6) of reviewer 1. Preparing these models takes some

work, so for the sake of reproducibility we also provide the final structures exactly as we used them for the calculations.

b) This is only a suggestion, so it remains for the authors to consider: I would suggest estimating the error from this atom cutoff, i.e., comparing the impact of 2k atom cutoff against that of full protein (or 15k atom cutoff). Note that this question comes naturally (at least to me) when I read the third paragraph of the Section SQM2.20 scoring (last of page 3). Then it occurred to me that the ligand set contains charged molecules, and electrostatics are long-range interactions...

We followed the suggestion and rescored the PL-REX dataset using the complete protein to compare the energy difference to our default model with the cutoff of 10 Å from the active site. The results show that an error using a smaller part of the protein is negligible in individual systems with a maximum difference of 0.05 (in the 02-HIV-PR target). The averaged R^2 resulted in the same value as when 10 Å cutoff is used and the worst case changed only by 0.01. This shows the interaction and solvation energies are already converged at the 10 Å cutoff. There is, however, a significant difference in computational time of such calculations which fully justifies the use of the smaller model. The average time of one SQM2.20 score calculation grows from 20.3 minutes in the 10 Å model to 74.2 minutes for the whole protein.

We have added these results in the new Section S6 of the Supplementary Information and refer to them by adding the following sentence in the Introduction on pg. 2 (new text underlined):

“For efficiency, the score is evaluated on a model of ~2,000 atoms, comprising all residues within 10 Å around all the overlaid ligands in each target protein. We had verified that this model perfectly reproduces the computationally more demanding scoring in a whole protein (see the SI, Section S6).”

9) In the last paragraph of the introduction it is mentioned that SQM2.20 is meant to fill the gap between very fast but inaccurate SFs and high-level DFT calculations where additional system constraints are needed. However, I see a very shy mentioning of timing information regarding SQM2.20 in the text. I believe such information should be duly pointed in the main text, already in the introduction, as indicated at the beginning of this point.

a) I would suggest adding a plot with exact calculation times for PM6-D3H4X and systems with ~2k atoms. The plots I would make are according to protein, showing the calculation times for each complex. This is just a suggestion though. Alternatively, consider adding more statistical descriptors (max, min, ...). Note that you can add this data to the SI with respective mentioning where it can be found. If you just say in the main text that the average calculation time is 10 minutes, that is great and more precise than “a few minutes”. I know some people would criticize saying it takes too long. But you show the significant gain in accuracy offered by your SF. And like you mention, SQM2.20 is supposed to fill in the gap between high-level and low-level. Therefore, it is supposed to be applied to 100-1000 structures. 5 minutes waiting time is perfectly fine. (5 minutes is just an example).

We agree that the timing of the calculations needs clarification. We added a more detailed discussion of the timing, which is described in our response to the previous point. Also, we added additional text to the SI Section S5 to clearly distinguish between the timing of the

overall score, and the timing of the interaction energy calculations discussed there in the context of comparison to the DFT calculations. The newly added Section S6 of the Supplementary Information includes both a table listing also the minimum and maximum values as suggested, and a plot of the computation time as a function of the number of atoms in the system (Figure S4).

To clarify the amount of time needed for scoring of one ligand, we added the following sentence into the Introduction (new text underlined):

“The SQM2.20 SF, with an average calculation time of ~20 minutes per P-L complex, is intended to fill the gap between very fast but less accurate SFs used in docking or virtual screening and much more expensive quantum chemical methods such as density functional theory (DFT).”

And we repeat it in the conclusions where it was appropriate:

“Conventional SFs and ML-based approaches are of course much faster (taking mere seconds to complete), and SQM scoring with an average computational time of about 20 minutes is not their direct replacement (For details on the timing, see SI, Section S6). “

b) From a psychological point of view, the 643 hours required by DFT on average is a lot. I don't need to think any further to be scared and immediately demotivated. But 0.05 hours mean nothing, and I am “emotionless” towards SQM2.20's efficiency. I would suggest putting in parentheses the times in minutes. Just a psychological thing.

We have modified the Table S3 in the S5 Section of the Supplementary Information to show the CPU time for PM6-D3H4X in minutes; we agree that this is more illustrative.

10) It is also mentioned at the end of introduction that SQM2.20 may be used at any stage of CADD. However, as far as I understood, there is no interface provided. So, the users would have to implement their own workflow? Is it offered as a service? I would recommend clarifying this point, making special reference if and when the authors plan to provide an interface. If something will be available from CUBY (to protect conflict of interests), it would be interesting to know. It is also OK to simply mention that implementation and adoption of the workflow to each user's environment is straightforward.

This work should be considered as a pilot study demonstrating the performance of the proposed SF and describing the computational protocol. The computations that make up the SF can be performed using freely available software packages that are referenced in the manuscript, enabling the readers to implement it themselves. We are currently working on automating and integrating the entire workflow into a more user-friendly package, but this is not the subject of this work, both because it is still under development and because it will contain additional protected intellectual property. For these reasons, we are adding only a small note in the manuscript that such an implementation is being developed, and we will announce it separately once available. We added the following sentence to the very end of the Conclusions where possible applications of the SF are mentioned:

“... To facilitate this, we plan to implement the SQM2.20 scoring function in a standalone tool that would automate the entire workflow; these developments will be announced separately in the future.”

11) A medicinal chemist that might be interested in using SQM2.20 would ask for the structures of the ligands for each protein. A 2D representation like that provided by Chem Draw or Marvin Sketch (or anything equivalent) would be nice. If possible, choosing a representation that reflects the orientation of the ligand inside the binding pocket for each set, or a representation faithful enough of the relative binding poses.

We added a new part of the Supplementary information with 2D structures of the ligands. A reference to this part of the SI was added to the “Methods/PL-REX dataset” Section of the paper:

“2D structures of all ligands, as well as schematic drawings of the binding mode in the representative P-L complexes, are provided in the Part 3 of the Supplementary Information.”

12) PL-REX contains metalloproteins with Zinc. If I am not mistaken, in previous work of the authors, it is mentioned that PM6-D3H4X has difficulties in treating the Zinc atoms. It is also mentioned that DFTB3, parametrization 3OB, offers a much more robust evaluation of scoring when Zinc is present. Also, in this publication (10.1021/acs.jcim.9b01171), you show how great DFTB3 is. You invested in the development of the H5 correction for DFTB3, which is newer than D3H4X. This is particularly interesting to analyse further.

We answer this together with point b) below.

a) Since these publications (present and 10.1021/acs.jcim.9b01171) discuss similar topics, maybe a few comments regarding how the overall results compare are possible?

The PLFrag dataset introduced in 10.1021/acs.jcim.9b01171 also covers protein-ligand complexes, but its purpose is different. It features mainly benchmark gas-phase DLPNO-CCSD(T) interaction energies in smaller fragments of the P-L complexes, making it a tool for the validation and development of methods for calculating the same quantity. This work is one of the arguments supporting the choice of PM6-D3H4X for the calculation of the interaction energy in SQM2.20 SF, as this method performed well there, and we cite the paper (as ref. 18) at the point where we introduce PM6-D3H4. We have to admit that this reference is not introduced very well in the text because it refers to unspecified “large systems” instead to P-L complexes, and it also misses the argument of favorable timing compared to other SQM methods (which we discuss in point b) below), and we thus change the sentence from:

“In our experience, this method provides the most accurate description of non-covalent interactions in large systems.¹⁸”

to:

“In our experience, this method provides the most accurate description of non-covalent interactions in large systems including fragments of protein-ligand complexes.¹⁸ Another reason for choosing PM6-D3H4X is the linear-scaling implementation of PM6

in MOPAC,¹⁹ the MOZYME algorithm,¹² which provides a significant speedup compared to other SQM methods.”

The PLFrag dataset, however, does not provide a benchmark for validating the complete scoring function - it was built from fifteen complexes of different targets, each with a single ligand, so there are no series of ligands the SF would be applicable to.

b) I am curious to read a comment from the authors on PM6 Vs. DFTB. Did the authors abandon DFTB3 as alternative for scoring? Note that this does not necessarily imply any change in the main text because DFTB is obviously not the subject and is completely disregarded.

Our preference for PM6-D3H4X is not contradicting our previous work, we used other methods including DFTB3 only when necessary. In small model systems, PM6-D3H4X and DFTB3 yield similar accuracy, but PM6 becomes much more efficient in larger ones - this was addressed in the above point. Furthermore, the COSMO2 solvent model for PM6 is more accurate (and also more computationally efficient) than solvent models available for DFTB3.

The zinc metalloproteins mentioned above were a specific case where DFTB3 addressed problems of an earlier version of PM6-based scoring. However, with the improvements included in SQM2.20, special treatment of zinc is no longer needed. The PL-REX dataset also contains two Zn metalloproteins (01-CA2 and 10-MMP12) and we show that SQM2.20 works well for both. The reason is the use of the COSMO2 solvent, this was already shown for the example of CA2 in our paper introducing COSMO2 [already cited in the manuscript; Rezac, JCIIM (2019), 59 (1), 229]. This is illustrated by the difference between the SF using a default PM6-D3H4/COSMO and SQM2.20 which uses COSMO2, with correlations with R^2 of 0.41 and 0.67, respectively. To highlight this improvement over the previous methods, we are adding a following note at the place where these two variants of the score are discussed, also referencing our previous work where we encountered the problems with zinc:

“Also, the transition from COSMO to COSMO2 significantly improves the description of 01-CA2, a zinc metalloprotein, as it remedies an issue we previously observed in this class of systems.²⁹”

c) Did the authors try to use SQM2.20 on the datasets offered in 10.1021/acs.jcim.9b01171. Is the correlation still as good? This does not have to involve any change in text or I don't expect any additional calculations regarding this subject.

This overlaps with point a) above, we provide a complete answer there.

13) The plots in SI, Section S2 show the comparison between SQM2.20 score and the experimental Gibbs free energies of binding, for all targets. As far as the description goes in SI, not all systems have K_i documented. Some seem to have exclusively IC_{50} . Though IC_{50} is proportional to K_i , the proportionality constant is not always 1. Further, K_i is an equilibrium constant in the presence of a substrate. In my opinion, a true thermodynamic comparison between the SQM2.20 scoring, and the K_i requires including the substrate, which was never mentioned anywhere in the text (I assume it was fully neglected). In my interpretation, DG_{bind} refers to the reaction $P + I = PI$, meaning that it relates to K_d . Surely K_i also reflects binding,

but the impact of the natural substrate should be made explicit in the thermodynamic expressions.

The reviewer is right in that the comparison of scoring to biochemical/biophysical experimental data deserves a more thorough explanation. The inhibition constant K_i is in principle equivalent to the true thermodynamic equilibrium dissociation constant K_d , as long as it concerns competitive inhibition (that is the case here). The value of K_i is extrapolated from measurements at different substrate concentrations, and is thus independent of it. In the literature on benchmarking calculations on experimental data, these variables are often treated the same, and we do not make an exception here. Also, we explain how we handle IC_{50} values and why we consider them valid in the present application. We added a new paragraph to the Methods Section on these topics:

“For the majority of the complexes, experimental dissociation constants (K_d) or inhibition constants (K_i) are available, which we treat as being equivalent under the assumption of a competitive inhibition mechanism. In the remaining cases, we use experimental IC_{50} values to approximate the K_i as $IC_{50}/2$, assuming that the concentration of the substrate in the experiment is close to the Michaelis-Menten constant. Even if this approximation was not accurate, it is a linear relationship that would not affect the correlation between scores and the experiment, as long as all the IC_{50} values come from a consistent series of experiments under the same conditions, which we verified in the original sources. Finally, we convert the experimental K_d , K_i or its estimate to a free energy of binding (ΔG_{bind}^0) that is compared to the calculations. It should be kept in mind that in the case of the IC_{50} values, it is only an estimate, but because this approximation does not affect the final results, we do not mention it explicitly in the remainder of the paper.”

a) In this sense I believe it is thermodynamically incorrect to call the experimental affinities “experimental ΔG_{bind} ” in Figure S1. In particular, the substrate will also have binding, entropy, proton-exchange contributions that will affect binding of the inhibitor. Furthermore, IC_{50} is to be interpreted as a concentration, not an equilibrium constant. pK_i or pIC_{50} are good enough.

We discuss this in the response to the previous point, and we added a note to the paper that in the case of IC_{50} values, the resulting ΔG_{bind} is only an estimate, but one which is valid for the data we are using.

14)

a) What is the criterion for “confidence” used for generating PL structures for ligands with no crystal structure?

We built the models only when we were confident to do so, which can be summarized with the following set of rules: We have only added / changed small functional groups to ligands with experimental structure, without changing the core of the parent ligand. These modifications have to fit into the binding sites without clashes, have a small number of rotatable bonds (leading to a small number of possible conformations). We added a definition of all the rules used for constructing the PL-REX dataset to the Section S1 of the SI, with the following sentence on modeling the ligands:

“We have only added small functional groups that fit into the binding sites without clashes, have a small number of rotatable bonds (i.e. a small number of possible conformations) and retain the core of the parent ligand.”

And we added the following reference to the particular section of SI on Page 4 of the manuscript.

“(following the rules listed in the SI, Section S1)”

15)

a) Schrodinger’s protein preparation wizard determines the protonation states of the ligand in the pocket. How do these compare to the ones proposed by the authors?

First of all, we need to emphasize here that we used a single representative protein structure for each target, into which all the other ligands were inserted by overlaying the proteins in all the crystal structures of complexes of each target. We thus determine the protonation of the protein first (considering all the ligands in the series), and then go back to the individual ligands and choose a protonation that is compatible with the protein environment. This was done manually, with special care, and it can not be achieved using automatic tools only. All non-trivial issues encountered in this process are listed in the S1 Section of SI (and are referred to in the main text). We improved the description of this protocol in the manuscript where the preparation of the systems is described, in the Results Section on pg 4 (new text underlined):

“A single representative protein structure per target was selected for scoring based on the criterion that it could best accommodate all the ligands (after the exclusion of incomplete structures). First, a single protonation state of the selected protein structure was determined with respect to the prior literature, experimental conditions and all the ligands in the series. Second, the protonation of ionizable groups in the ligands was solved and corrected manually according to the experimental conditions, pKa calculations, the literature and, most importantly, adjusted to match the selected protein structure. Non-trivial issues, i.e. the protonation states of 20 ligands altered upon binding (a proton is released in 11 cases and taken up in 9 cases) are listed in the SI, Section S1.”

We also clarified the first mention of the preparation of the structures on pg. 1 in the Introduction:

“Therefore, we compiled a unique dataset of reliable experimental structures and affinities, the Protein–Ligand Refined EXperiment (PL-REX) set. It comprises ten diverse protein targets, each with ten to thirty ligands, ~~with all the complexes meticulously prepared and manually checked for non-trivial issues.~~ Although the PL-REX dataset comprises multiple crystal structures of P–L complexes within each series, we chose a single protein conformation for each target, into which all the other ligands were inserted by overlapping the crystal structures. The protonation states of selected proteins and ligands were meticulously prepared and manually checked for non-trivial issues.”

None of the automatic protocols, including the Protein Protonation Wizard, will yield similarly consistent results, as they do not consider the protein structure along with all the ligands at the same time. A typical case, where an automated procedure fails, is HIV-PR, where two active-site aspartates are close to one or more atoms of the ligand. Because these contact distances fall within any reasonable cavity radius, the carboxylates are not subject to being neutralized in the automatic process, and will both be represented as negatively charged. However, when the ligand interacts with the aspartates via a hydroxyl or similar neutral group, one of the aspartates has to be modeled as neutral. In PL-REX, the OD1 “lower” oxygen of Asp25’ of HIV-PR was modeled correctly as protonated. Moreover, the protonation states of the 5HVP ligand might differ here as well. The pepstatin-based inhibitor interacts here with Asp29 of the protein and to have a single representative protein structure for all the ligands, we considered the proton transfer to the pepstatin instead of to the protein, which is then compensated by the protonation penalty.

16)

a) Though clear for me, I think it should be stated in page 9 why PM7 fails in the scoring. This will help unexperienced users choose their quantum chemical method.

To address this point, we have added the following note on the performance of PM7 as suggested, with a reference to our earlier works where this is analyzed in detail:

“We also note that PM7/COSMO²⁸ resulted in a slightly decreased average correlation (by 0.10) and that it failed to produce reasonable correlation in three targets (03-CK2, 09-CDK2, 10-MMP12, Table S1). This is not a surprising result, since PM7 tends to overestimate interaction energy in large systems in general,^{18,29} which also affects its ability to describe protein-ligand complexes.¹⁸”

17)

a) A diagram with workflow would make everything clearer from the protocol point of view.

We added a diagram outlining the whole protocol, it is now Figure 1 in the manuscript. A reference to the figure was added to the main text at the end of the paragraph introducing the components of the SF in the Introduction:

“The entire workflow, from preparing the structures for calculation to evaluating each component of the SQM2.20 score, is outlined in Figure 1.”

18)

a) I would show plots in Figure S1 for other SFs. I suppose these were previously prepared, so they should simply be included for transparency.

We prepared the plots for the other scoring functions and added them as a second part of the Supplementary Information. In that document, we also mention that the data used to generate these plots are available in the repository associated with the paper. A reference to the Supporting Information has been added to Methods where these SFs are introduced:

“The scores obtained with these methods are available in the PL-REX repository, and their plots against the experimental results are provided in Part 2 of the Supplementary Information.”

19)

a) I suppose that calculations are ran on charged systems, i.e., no charge neutralization is used. This should be explicit.

Yes, we are not altering the charge of the model. With the large 10 Å cutoff used to define the model, this is not necessary. First, any charges introduced at the boundary are far enough from the ligands. Second, they are exposed to the solvent which screens them effectively. The comparison with the calculations in a complete protein, discussed above in point 8), proves that this approach is safe.

However, we agree that this should be explicitly stated in the paper. We have added the following sentence in Methods/PL-REX dataset Section:

“Truncating the active site model could expose charged amino acid residues at the boundary; we do not neutralize these as they are far from the ligands and are effectively screened by the solvent model.”

20) The work of Chan and coworkers used for the quick estimation of entropy effects clearly shows that vibrational contributions to the GFN2-xTB Svib are not negligible. Further, neglecting these should have stronger impact on the thermodynamics of binding than Sconfig. I would be interested in comments/thoughts on this. Note that this does not necessarily require changes in the manuscript.

We are aware of the studies that highlight the importance of vibrational entropy. However, there is no practical method for calculating this contribution that would be applicable as part of an efficient scoring function. First, there are theoretical problems in separating the conformational part from low frequency vibrations. Second, even if a simple model based on the calculation of harmonic vibrations in the minimum structure were to be adopted, it would be very expensive, not only because of the size of the systems, but also because it would place additional demands on the numerical accuracy of the calculations if the low-frequency vibrations, which are the most important ones, were to be calculated reliably.

Of course, it would be necessary to include these (and other) terms if we wanted to estimate the absolute binding free energies, but that is not the goal of our SF. Also, because of the problems mentioned above, we believe that methods based on MD simulations address this problem better than attempts to isolate Svib and calculate it using additional, often severe approximations. In the present SF, we neglect this contribution and rely on the assumption that part of it cancels out (being the same for all ligands in the series) and that another part does not affect the correlation (being inversely proportional to the strength of the interaction, which is already calculated). The remaining contribution is of course missing, and we include it in all the dynamical effects that we do not cover (as stated in the manuscript).

21) At the end of the results/discussion, it is mentioned that "Furthermore, as a non-empirical SF consisting of physically well-defined terms, it provides insight into the nature of P–L interactions, which may guide the rational design of better ligands."

a) There is no discussion regarding this subject in this manuscript, as to how the nature of PL interactions as described by the SQM calculations could guide rational ligand design. I would even claim that if I were to take 2 ligands and put them inside the protein, there is nothing in SQM2.20 telling me why one ligand is better than the other: the introduction of a fluorine leads to better dispersion, or electrostatics? There is only one singular value, the scoring, that predicts whether binding is improved. There is no rationale behind it, except potential user interpretation which, for inexperienced users may be biased by wishful thinking. I would either suggest removing this sentence, as it is misleading with regard to SQM2.20's scope or make reference to quantum chemical approaches that can cover such guidance scenarios.

We agree that this is not discussed in the present work, and we thus remove the particular sentence from the manuscript. (To provide a full explanation, this note referred to some of our previous studies where earlier versions of the SQM SF were used in more complex scenarios to provide such insight.)

22)

a) GlideSP offers better results than GlideXP. Maybe a sentence to comment on this? Even if it is just saying that you noted this, but there is no clear reason as to why since you don't have the source code nor developed any of the methods. Anyhow, it could bring awareness regarding the need for more trustworthy scoring functions, like the one you are proposing.

We have noticed the worse performance of GlideXP-min (with an average correlation R^2 of 0.32) compared to GlideSP-min (with R^2 of 0.40). We double-checked the correctness of our calculations and found no issues. This, moreover, seems to be a general trend observed also by others - the same was found in the recent CASF benchmark study (see *J. Chem. Inf. Model.* 2019, 59, 2, 895–913, already cited in the paper).

Nevertheless, this is an interesting result and it deserves mentioning in the paper - we added a comment on this into the main text of the manuscript on pg 8 and we also repeat the citation to the CASF study there:

"Here we found one result worth noting – the more sophisticated GlideXP scoring function did not outperform the GlideSP; however, this is consistent with an earlier benchmark study.²¹"

Reviewers' Comments:

Reviewer #2:

Remarks to the Author:

The authors have sufficiently responded to my suggestions.

Nevertheless, small incremental progress over the same authors previous manuscripts, lack of user interface that would enable inexperienced user to use the presented methodology (called "pilot study") limit the readership significantly.

Therefore, I remain of the opinion that the manuscript is not suitable for general-audience journal and should be transferred to a specialized field journal.

Reviewer #3:

Remarks to the Author:

I was asked to review the paper by Řezáč and co-workers, and most specifically, whether the concerns of Reviewer #1 have been addressed in the revisions. Therefore, my report will mostly reflect this Task.

Reviewer #1 focused mostly on points that would allow for a clarification of not only the scientific language/terms but also the reproducibility of the results. I can say that I agree with the generality of the points raised before. Overall, I think the authors appropriately revised the manuscript, competently tackling the points raised. I must also add that Reviewer #2 did a pretty good job in the scrutiny of the paper which allowed for further clarification of the manuscript, where the reproducibility questions were also an issue, but the authors fully complied with his/her concerns regardless of the difficulty and extra work required.

Overall, the report of the new SQM2.20 that outperforms 22 scoring functions (along with MM and DFT), while providing the results in the scale of minutes per complex, is indeed important and merits its publication. The availability of the PL-REX benchmark is also a plus of this work.